# Towards Graph Foundation Models: A Study on the Generalization of Positional and Structural Encodings

**Billy Joe Franks**[*]
*University of Kaiserslautern-Landau*

*billy.franks@rptu.de*

**Moshe Eliasof**[*]
*University of Cambridge*

**Semih Cantürk**
*Université de Montréal*
*Mila - Quebec AI Institue*

**Guy Wolf**
*Université de Montréal*
*Mila - Quebec AI Institue*

**Carola-Bibiane Schönlieb**
*University of Cambridge*

**Sophie Fellenz**
*University of Kaiserslautern-Landau*

**Marius Kloft**
*University of Kaiserslautern-Landau*

**Reviewed on OpenReview:** *https://openreview.net/forum?id=mSoDRZXsqj*

## Abstract

Recent advances in integrating positional and structural encodings (PSEs) into graph neural networks (GNNs) have significantly enhanced their performance across various graph learning tasks. However, the general applicability of these encodings and their potential to serve as foundational representations for graphs remain uncertain. This paper investigates the fine-tuning efficiency, scalability with sample size, and generalization capability of learnable PSEs across diverse graph datasets. Specifically, we evaluate their potential as universal pre-trained models that can be easily adapted to new tasks with minimal fine-tuning and limited data. Furthermore, we assess the expressivity of the learned representations, particularly, when used to augment downstream GNNs. We demonstrate through extensive benchmarking and empirical analysis that PSEs generally enhance downstream models. However, some datasets may require specific PSE-augmentations to achieve optimal performance. Nevertheless, our findings highlight their significant potential to become integral components of future graph foundation models. We provide new insights into the strengths and limitations of PSEs, contributing to the broader discourse on foundation models in graph learning.

## 1 Introduction

In recent years, graph neural networks (GNNs) and machine learning on graphs have gained significant attention due to their versatility in processing complex data across various domains, including biology,

---

[*]Equal contribution.

social networks, chemistry, and recommendation systems. Comprehensive surveys on these advancements are available in Bronstein et al. (2021) and Waikhom and Patgiri (2023). GNNs are fundamentally designed to extend traditional deep learning techniques to graph-structured data by aggregating information from local neighborhoods, thereby enabling the extraction of structural and relational features intrinsic to graphs. Despite considerable progress, these methods continue to face several challenges, including issues related to expressivity, such as the ability to distinguish between non-isomorphic graphs (Morris et al., 2019) and substructure counting (Bevilacqua et al., 2022), the generalization of models across different graph datasets and tasks (Ju et al., 2023), as well as the connection between generalization and expressivity that has been studied recently (Li et al., 2024).

In terms of expressivity, i.e., the ability of GNNs to approximate functions (Huang and Villar, 2021), message-passing neural networks (MPNNs) are known to have limitations that can hinder their performance in practical applications (Morris et al., 2019; Xu et al., 2019; Morris et al., 2023; Zhang et al., 2021). To address these limitations, various strategies have been proposed, with two prominent approaches focusing on augmenting the input node features of a graph with positional and structural encodings (PE and SE, respectively). One such approach involves using random node features (RNF) (Abboud et al., 2021; Sato et al., 2021), which have been shown to enhance the expressivity of MPNNs by increasing their ability to distinguish between nodes. However, despite the theoretical benefits, RNF has not consistently improved performance on real-world datasets (Bevilacqua et al., 2022; Bechler-Speicher et al., 2025). To address this inconsistency, recent studies have demonstrated that learned PEs (Eliasof et al., 2023) and learned graph normalization techniques (Eliasof et al., 2024) can leverage RNF to improve both expressivity and downstream task performance. Another approach to overcoming the expressivity limitations involves incorporating spectral PEs, which are often derived from a partial eigendecomposition of the graph-Laplacian matrix (Dwivedi and Bresson, 2021; Kreuzer et al., 2021; Wang et al., 2022a; Lim et al., 2023; Rampášek et al., 2022). Additionally, structural encodings based on random walks (Dwivedi et al., 2022a) have also proven to be beneficial. More recently, the graph positional and structural encoder (GPSE) (Cantürk et al., 2024) introduced a unified framework for learning from both positional and structural encodings while utilizing RNF. Overall, these methods have shown improvements, particularly on specific graph benchmarks where models are trained from scratch for each task. Notably, GPSE also demonstrated some degree of generalization to new tasks.

Simultaneously with the advancements in GNNs, foundation models (FM) have emerged and revolutionized the fields of natural language processing (NLP) and computer vision (CV). These developments present an exciting opportunity to bring similar transformative advances to machine learning on graphs. In NLP, models like BERT (Devlin et al., 2019) and GPT (Brown et al., 2020) have demonstrated how pre-training on large datasets can produce models that generalize effectively to a variety of downstream tasks with minimal to no fine-tuning. Inspired by these successes, recent works (Liu et al., 2023; Galkin et al., 2024; Huang et al., 2023a; Mao et al., 2024; Shi et al., 2024; Xia et al., 2024; Liu et al., 2024) have begun exploring the creation of graph foundation models (GFM), which could offer similar benefits for graph-based tasks. Such models would not only simplify the training process across multiple tasks but also significantly reduce the computational resources, energy consumption, and data requirements typically associated with task-specific training. This aligns with the broader objective of developing more environmentally sustainable machine learning practices.

However, building full-scale foundation models for graphs presents unique challenges. Unlike textual or visual data, which often exhibit relatively uniform structures, such as 2D grids with fixed resolutions in images, graphs can vary widely in their topology, size, and complexity. Additionally, the input feature space for graphs is highly heterogeneous; while pre-trained models for images typically operate on a common input space (e.g., RGB values), different graph datasets often have vastly different input features, ranging from molecular atom types to paper attributes in citation networks. This variability across datasets complicates the design of a universal, pre-trained graph model. Given these challenges, while diverse input features remain an open issue, a promising direction is to focus on learning foundational graph representations that capture key structural properties inherent in graph data. In particular, this paper emphasizes the role of positional and structural encodings (PSEs), which enhance node identifiability within a graph and offer a more practical approach to developing versatile graph encoders that can generalize across various tasks and datasets. This work is inspired by the growing body of research on pre-trained encodings for graphs,

particularly the recent work on the graph positional and structural encoder (GPSE), which introduces a graph encoder specifically designed to capture diverse PSEs.

**Our Contributions.** In this paper, we both practically and theoretically assess whether PSEs can function as foundation models or, alternatively, as a building block for future foundation models. We especially focus on the recent GPSE model which we consider the state of the art when it comes to PSEs. We note that, our main contribution is the study of the role of GPSE as a building block within future GFMs, rather than proposing a new methodology. We believe that these contributions are useful and important for the graph-learning community. To be precise we:

- Discuss the expressivity of GPSE as a standalone model, as well as general PSEs contribution to expressivity when they are used to augment downstream GNN models (Section 4).

- Show that GPSE accelerates convergence to better performance. Specifically, it achieves strong results more quickly due to its pre-training phase (Section 5.1).

- Determine whether sample size affects model performance when using GPSE and other PSEs, providing insights into their effectiveness in data-scarce settings (Section 5.2).

- Analyze and compare various PSE's performance across different datasets to assess their generalization capability. The evaluation indicates that GPSE seems to perform generally best, while some failure cases still remain (Section 5.3).

Our findings indicate that while GPSE and other PSEs do not yet function as standalone GFMs due to their performance being somewhat dependent on the dataset in question, they show potential as valuable components in future GFMs due to their increased robustness in data-scarce settings and their improved generalization. This potential is primarily attributed to GPSE's strong generalization capabilities, which we identify as a critical factor for enhancing downstream performance (Section 5.3). These results underscore the importance of studying generalization in graph learning, a key area for advancing the development and understanding of GFMs, as also highlighted by Morris et al. (2024).

## 2 Related Work

This work studies the connection between GFMs, PSEs, and expressivity, in GNNs. We now discuss related work on these topics.

**Graph foundation models.** Inspired by the success of FMs in NLP (Brown et al., 2020) and CV (Dosovitskiy et al., 2021), there has been increasing interest in developing similar models for graph-based tasks. These GFMs aim to generalize across various graph-related tasks by leveraging large-scale pre-training and adaptable architectures. Liu et al. (2023) provide a comprehensive analysis of the current state of GFMs. Their study highlights significant challenges in the development of general GFMs, noting that the diversity of graph structures and tasks makes it difficult to create models with broad applicability. Shi et al. (2024) emphasize the potential impact of GFMs on a wide range of applications, from social network analysis to molecular modeling. Also, Mao et al. (2024) discuss the potential directions for designing effective GFMs, and propose a taxonomy of task-specific GFMs, outlining the key design principles required to achieve generalization across various graph tasks. Xia et al. (2024); Liu et al. (2024), and Huang et al. (2023a) propose GFMs by use of graph-tokenization, text-attributed graphs, and in-context learning for graphs. These models have difficulties with balancing task specificity with generalization capabilities or regression tasks. A promising example of a domain-specific GFM is presented in Galkin et al. (2024), a GFM tailored for knowledge graph tasks was shown to be successful, excelling in tasks such as knowledge graph completion and link prediction, showcasing how specialized GFMs can achieve state-of-the-art results in specific domains. However, there is still a need to develop GFMs for other domains like biology and chemistry, where PSEs are commonly applied, as recently discussed in Frasca et al. (2024).

On a different note, recently, there has been a series of works that utilize large language models (LLMs) as FMs for graph learning tasks, such as in Fatemi et al. (2024); Perozzi et al. (2024) and others. We note

that, this direction is very promising in terms of the large availability of pre-trained models, as well as the abundance of textual data used in LLMs, compared with the scarcity of graph data. However, in this paper, we focus on employing GNNs and concepts within, to study whether and how they can be oriented towards GFMs, which can be later combined with LLM approaches.

**Positional and structural encodings.** PSEs are features that provide information on the graph's structure and node positions. The seminal work in Vaswani et al. (2017) introduced the concept of positional encodings in the transformer architecture, demonstrating their effectiveness in capturing the order of sequences. In graph learning, the inclusion of PSEs was shown to be significant for enhancing the performance of models in various tasks (Srinivasan and Ribeiro, 2020), as we now discuss. Methods utilizing graph Laplacian eigenvectors as PEs (LapPE) have been explored by Kreuzer et al. (2021) and Rampášek et al. (2022). Kreuzer et al. (2021) also investigated the use of electrostatic potential encodings, providing a novel approach to structural representation. Ying et al. (2021) proposed using shortest paths as encodings, which have shown promise in capturing graph structures. Shiv and Quirk (2019) introduced tree-based encodings, offering a unique perspective on hierarchical data representation. Random walk-based encodings (RWSE) have been explored by Rampášek et al. (2022); Dwivedi and Bresson (2021) and Dwivedi et al. (2022b), highlighting their versatility in various applications. The use of heat kernels for SEs has been discussed by Kreuzer et al. (2021) and Mialon et al. (2021). Subgraph-based encodings, which have been effective in capturing local graph structures, were the focus of Zhao et al. (2022); Bouritsas et al. (2022); Chen et al. (2022) and Yan et al. (2023). Additionally, Ying et al. (2021) considered node degree as a simple yet powerful SEs. Furthermore, a different set of works has shown that RNF (Abboud et al., 2021; Sato et al., 2021; Franks et al., 2023) are theoretically strong PEs, and they were further utilized in Eliasof et al. (2023) for learnable PEs via their propagation. Recently, it was shown in Cantürk et al. (2024) that RNF can be combined with the aforementioned PSEs to learn a general encoding, called graph positional and structural encoding (GPSE). While PSEs have shown great promise in improving performance on various tasks, their supposed effects on GFMs has been under-explored, with preliminary experimental results shown in Frasca et al. (2024). In this paper, we offer a detailed study on the generalization ability of learned PSEs, accompanied by a theoretical discussion of their expressiveness.

**Expressivity of GNNs** The expressivity of GNNs has been widely studied in recent years (Xu et al., 2019; Morris et al., 2019; Maron et al., 2019a), highlighting the theoretical limits of GNNs. To improve the expressivity of GNNs, several approaches were proposed and studied, from substructure encodings (Bouritsas et al., 2022), subgraph GNNs (Bevilacqua et al., 2022; Zhao et al., 2022; Bevilacqua et al., 2024), to homomorphism counting (Zhang et al., 2024), to the augmentation of input node features with PSEs such as RNF (Dasoulas et al., 2020; Murphy et al., 2019; Sato et al., 2021; Abboud et al., 2021; Franks et al., 2023). In this work, following GPSE that utilizes RNF as part of its architecture, we theoretically discuss the expressive power of GPSE, and study whether RNF are required in practice for downstream tasks when utilizing a framework such as GPSE. A further discussion of expressivity not considered here can be found in Appendix A.

## 3 Notations and Background

Throughout this paper, we consider (undirected) graphs. An undirected graph $G$ is defined by the pair $(V(G), E(G))$ with *finite* sets of nodes $V(G)$, and edges $E(G) \subseteq \{\{u, v\} | u, v \in V(G)\}$. W.l.o.g. we assume that $V(G) = \{1, \ldots, n\}$. Within machine learning on graphs, nodes typically carry feature vectors which lead to undirected labeled graphs $G = (V(G), E(G), \ell)$, where $\ell : V(G) \to \mathbb{R}^d$ assigns each node a feature vector.

**Color refinement (CR)** CR is a well-studied heuristic for the graph isomorphism problem, originally proposed by Leman and Weisfeiler (1968). Intuitively, the algorithm determines if two graphs are non-isomorphic by iteratively coloring or labeling vertices. Formally, let $G = (V(G), E(G), \ell)$ be a labeled graph. In each iteration, $t > 0$, CR computes a vertex coloring $C_t^1 : V(G) \to \mathbb{N}$, depending on the coloring of the neighbors. That is, in iteration $t > 0$, we set

$$C_t^1(v) \coloneqq \mathsf{RELABEL}\Big(\big(C_{t-1}^1(v), \{\!\{C_{t-1}^1(u) \mid u \in N(v)\}\!\}\big)\Big),$$

for all vertices $v \in V(G)$, where RELABEL injectively maps the above pair to a unique natural number, which has not been used in previous iterations. In iteration 0, the coloring $C_0^1 := \ell$ is used, which can be converted to a natural number in a multitude of ways. To test whether two graphs $G$ and $H$ are non-isomorphic, we run the above algorithm in "parallel" on both graphs. If the two graphs have a different number of vertices colored $c \in \mathbb{N}$ at some iteration, CR distinguishes the graphs as non-isomorphic. Moreover, if the number of colors between two iterations, $t$ and $(t+1)$, does not change, i.e., the cardinalities of the images of $C_t^1$ and $C_{i+t}^1$ are equal, the algorithm terminates. For such $t$, we define the stable coloring as $C_\infty^1(v) = C_t^1(v)$, for $v \in V(G \,\dot\cup\, H)$.

**Message-passing neural networks (MPNNs)** Intuitively, MPNNs learn a vectorial representation, i.e., a $d$-dimensional real-valued vector, representing each node in a graph by aggregating information from neighboring vertices. Formally, let $G = (V(G), E(G), \ell)$ be a labeled graph with initial node features $\ell(v) =: \boldsymbol{h}_v^{(0)} \in \mathbb{R}^d$. A typical MPNN architecture is obtained by stacking permutation-equivariant parameterized neural layers. Following, Scarselli et al. (2008) and Gilmer et al. (2017), in each layer, $l \geq 0$, we define the updated node features as:

$$\boldsymbol{h}_v^{(l+1)} := \mathsf{UPD}^{(l+1)}\Big(\boldsymbol{h}_v^{(l)}, \mathsf{AGG}^{(l+1)}\big(\{\!\!\{ \boldsymbol{h}_u^{(l)} \mid u \in N(v) \}\!\!\}\big)\Big) \in \mathbb{R}^d, \tag{1}$$

for each $v \in V(G)$. The functions $\mathsf{UPD}^{(l)}$ and $\mathsf{AGG}^{(l)}$ are typically learned. Two examples of common MPNN layers, that will be considered in this work, are the graph isomorphism network (GIN) layer (Xu et al., 2019) defined as:

$$\boldsymbol{h}_u^{(l+1)} = \mathrm{MLP}^{(l+1)}\left((1 + \epsilon^{(l+1)})\boldsymbol{h}_u^{(l)} + \sum_{\{u,v\} \in E(G)} \boldsymbol{h}_v^{(l)}\right), \tag{2}$$

where $\mathrm{MLP}^{(l+1)}$ is a multilayer perceptron (MLP) in layer $l+1$ and $\epsilon^{(l+1)} \in \mathbb{R}$ is a constant of layer $l+1$ differentiating a node from its neighbors. Further, the gated graph convolutional network (GatedGCN) layer (Bresson and Laurent, 2017) is defined as:

$$\boldsymbol{h}_v^{(l+1)} = \mathrm{ReLU}\left(U^{(l)}\boldsymbol{h}_v^{(l)} + \sum_{\{u,v\} \in E(G)} \eta_{uv}^{(l)} \odot V^{(l)}\boldsymbol{h}_v^{(l)}\right), \tag{3}$$

where $\eta_{uv}^{(l)} := \sigma(A^{(l)}\boldsymbol{h}_u^{(l)} + B^{(l)}\boldsymbol{h}_v^{(l)})$, $U^{(l)}, V^{(l)}, A^{(l)}, B^{(l)}$ are linear layer weight matrices, and $\sigma$ is the sigmoid activation.

### 3.1 Graph positional and structural encoder (GPSE)

GPSE (Cantürk et al., 2024) is a recently introduced GNN model that aims to learn and predict robust and transferable representations for positional and structural encodings using RNFs as inputs, and constitutes the main architecture studied in our paper. In this paper, we choose to work with GPSE because it generalizes other PSEs, such as the graph Laplacian eigenvectors or random-walk structural encodings, in the sense that it learns these PSEs during its training phase. In what follows, we provide a brief standalone introduction to the model; further information is available via the original paper.

**GPSE model.** GPSE employs an encoder-decoder architecture in which the encoder and decoder are trained jointly, whereas in inference, only the encoder is used. The encoder is a deep MPNN consisting of stacked GatedGCN (Bresson and Laurent, 2017) layers, with residual gating and skip-connections in-between layers. The decoder consists of multiple node-level MLP heads, each corresponding to a different PSE component.

The inputs to the model are graphs that are preprocessed to add a virtual node, and original node features replaced with 20-dimensional RNFs, $\boldsymbol{x}_i \sim \mathcal{N}(\mathbf{0}, \mathbf{I}) \in \mathbb{R}^{1 \times 20}$, which are first projected to match the inner dimension of the encoder, $d$ (512 in the original paper):

$$\boldsymbol{h}_i^{(0)} = \mathrm{ReLU}\Big(\boldsymbol{x}_i W_{\mathrm{inp}}\Big) \tag{4}$$

The resulting hidden representations are then passed through $L$ GatedGCN layers:

$$\boldsymbol{h}_i^{(l+1)} = \mathrm{ReLU}\left(\boldsymbol{h}_i^{(l)}W_1^{(l)} + \sum_{j\in\mathcal{N}(i)} \sigma\left(\boldsymbol{h}_i^{(l)}W_2^{(l)} + \boldsymbol{h}_j^{(l)}W_3^{(l)}\right) \odot \left(\boldsymbol{h}_j^{(l)}W_4^{(l)}\right)\right) \tag{5}$$

where $W_i^{(l)} \in \mathbb{R}^{d\times d}$ are learnable parameters for layer $l$, $\sigma$ is the sigmoid function, and $\odot$ represents element-wise multiplication. The output of the encoder is a latent representation $\boldsymbol{h}_i^{(L)}$, which is decoded at each head by a two-layer MLP to predict the respective PSE component (e.g. absolute value of the second eigenvector, or the fifth RWSE) for each node:

$$\hat{y}_{i,k} = \mathrm{ReLU}\left(\boldsymbol{h}_i^{(L)}W_{k,1}\right)W_{k,2} \tag{6}$$

with $W_{k,1} \in \mathbb{R}^{d\times d}$ and $W_{k,2} \in \mathbb{R}^{d\times 1}$ are learnable parameters.

GPSE additionally learns graph-level encodings in the absolute values of the graph Laplacian eigenvalues and cycle counts; to do so the authors sum the node-level representations to obtain graph-level ones, which are then similarly passed through a 2-layer MLP per PSE.

From here on we will assume that, given the structure of a graph $G = (V(G), E(G))$, namely without any labels, as well as some random features $X$, GPSE provides node embeddings $\mathrm{GPSE}(G, X)(v)$. Throughout most of this paper we consider GPSE to be pre-trained and thus fixed. We thus think of GPSE as a PSE that provides node embeddings.

**PSEs.** In this work, we will be comparing multiple PSEs, including GPSE. Here we give a brief overview of the most relevant PSEs. The Laplacian eigenvector positional encoding (LapPE) (Dwivedi et al., 2020; Lim et al., 2023) is the absolute value of the $l_2$ normalized eigenvector associated with non-trivial eigenvalues of the graph Laplacian. The random walk structural encoding (RWSE) (Dwivedi et al., 2022b) is the probability of returning back to the starting state of a random walk after $k$ steps. We refer to the combination of all PSEs used in Cantürk et al. (2024) as AllPSE. Lastly, we compare to Local-instance and Global-semantic Learning (GraphLog) (Xu et al., 2021), which constructs a locally smooth latent space by aligning the embeddings of correlated graphs/subgraphs. More details and some additional less relevant PSEs can be found in Appendix D,

**Downstream universality.** Throughout this work, we will be considering the expressivity that node augmentations (or positional and structural encodings) provide to GNNs. For example, as previously shown in Sato et al. (2021); Franks et al. (2023), random node features (RNF) added to GNN based on GIN layers enable the resulting model's universality. In this sense, RNF itself is not universal but we will refer to this as "RNF enables downstream universality". More generally, when we refer to the downstream universality or downstream expressivity of some PSE, then we are referring to the universality or expressivity that a GNN, that is able to approximate 1-WL, attains when its input features are augmented by this PSE.

## 4 The Expressivity of GPSE

In this section, we discuss the extent and relevance of expressivity for GPSE and other PSEs. We start by proving that GPSE using GatedGCN layers can universally approximate PSEs. We then discuss that PSEs do not enable downstream universality. In doing so, we also propose two additional GPSE-like models, that we call $\mathrm{GPSE}^-$, and $\mathrm{GPSE}^+$, that verify the statements we make.

### 4.1 GPSE is a Universal Node Encoder

Since GPSE is meant to be able to compute PSEs for any input graph and thus provide a powerful positional and structural node encoding, it is important to verify that GPSE is indeed a universal node encoder. Because some PSEs are harder—in a WL sense—to compute than CR, GPSE as an MPNN needs to be fully expressive. And indeed, GPSE is a universal node encoder, because the input graph is augmented with RNF, which we now show. First, note that GatedGCN layers can approximate GIN layers by the following theorem, which we prove in Appendix B.

**Theorem 1.** *Under mild assumptions on the input graphs, an MPNN consisting of sufficiently many GatedGCN layers can approximate an MPNN made up of GIN layers arbitrarily well.*

Combining Theorem 1 with Theorem 3 from Xu et al. (2019), which states that an MPNN made up of GIN layers can learn to approximate 1-WL, proves that GatedGCN can also simulate 1-WL. Furthermore, Theorem 4 from Franks et al. (2023) shows that, assuming sufficiently many layers and sufficient width of the layers within GPSE, any function on the graph structure (that is, functions of the form $f((V, E))$ ignoring node/edge features) can be computed. This proves that, indeed, under mild assumptions, GPSE is universal and can learn to embed any PSE.

### 4.2   PSEs do not Guarantee Downstream Universality

While GPSE can learn to embed any PSE, this is not sufficient to ensure the universality of the downstream model. In fact, PSEs in general do not provide universality of the downstream model. As shown in Cantürk et al. (2024), as well as in Table 1, a GIN downstream network augmented with GPSE is not able to accurately generalize on the EXP, CEXP, CSL, TRI, and TRIX datasets, that are often used as expressivity benchmarks (Abboud et al., 2021; Murphy et al., 2019). Let us assume that the pre-trained network in GPSE learned an identity mapping of the input random node features. Then, the resulting model is theoretically universal with high probability, as shown in Abboud et al. (2021) and Sato et al. (2021). This would imply that the issue lies in the pre-training of GPSE. That is, GPSE did not learn to encode PSEs which obtain high accuracy on some graphs that require high expressivity to distinguish.

**GPSE$^+$.** We empirically verify that by adding relevant "hard" graphs, i.e., graphs that require high expressivity, to the pre-training dataset of GPSE, GPSE is practically more expressive. Specifically, relevant hard graphs for the CSL dataset are 4-regular graphs. Thus we define GPSE$^+$ to be the exact same model as GPSE trained on MolPCBA ($>$300.000 unique graphs), as well as 1000 random 4-regular graphs, each with 24 nodes. We chose 24 nodes, as this is roughly the average number of nodes for MolPCBA. Indeed, Table 1 shows that, by adding these hard graphs to the pre-training stage, the performance of GPSE on CSL increases significantly. The performance on EXP also increases; however, since the added graphs are not particularly relevant to the EXP dataset, the performance does not change as dramatically as in the case of CSL. We also evaluated on TRI and TRIX, which are 3-regular graphs with triangles as labels, where again GPSE$^+$ outperforms GPSE. We note, that the use of 4-regular graphs was merely a demonstration focusing on the CSL dataset. In practice, either a task-tailored or a more general set of hard graphs would need to be used in the pretraining stage.

An interesting question that follows is, *If the pre-training of GPSE is done as in GPSE$^+$, is a GNN augmented with GPSE universal?* As stated before, if GPSE were to learn to output random features, the universality would be given for a sufficiently powerful downstream GNN. However, since GPSE is actually trained to embed PSEs and not to output random features, we can assume that the GPSE computed node-embeddings contain at most marginal amounts of randomness, regardless of whether a GPSE or a GPSE$^+$ model is used. Alternatively, if, instead, GPSE does not output random features, then this implies that GPSE will, at best, learn to output node embeddings that represent the orbit partition (not considering the input colors). Under this assumption, $\text{GPSE}(G, X)(v) = \text{GPSE}(G, X)(w)$ if and only if $v$ and $w$ are in the same orbit. However, under this assumption, there exist graphs $A$ and $B$ which a GPSE-augmented MPNN cannot distinguish.

Table 1: The inclusion of hard graphs in GPSE$^+$ pre-training is crucial to the model's success on expressivity datasets. Without RNF, GPSE$^-$ cannot perform well on these datasets. We report the accuracy (%)↑ of GPSE$^-$, GPSE, and GPSE$^+$.

| Method↓ \ Dataset→ | EXP | CEXP | CSL | TRI | TRIX |
|---|---|---|---|---|---|
| GPSE$^-$ | 49.8±0.3 | 71.9±2.9 | 9.3±3.3 | 74.8±1.1 | **95.1±0.0** |
| GPSE | 61.3±2.3 | 72.3±2.6 | 42.7±10.2 | 83.6±0.5 | 31.1±30.7 |
| GPSE$^+$ | **73.6±1.1** | **74.8±2.3** | **68.0±6.2** | **96.3±0.5** | 89.6±8.5 |

**Theorem 2.** *Let $f : G \to (V(G) \to \mathbb{D})$ be a function such that $f(G)(v) = f(G)(w)$ if $v$ and $w$ are in the same orbit of graph $G$. Then there exist colored graphs $A = (V_A, E_A, \ell_A)$ and $B = (V_B, E_B, \ell_B)$ such that the colored graphs $A' = (V_A, E_A, \ell'_A)$ and $B' = (V_B, E_B, \ell'_B)$ with $\ell'_A(v) := RELABEL(\ell_A, f(V_A, E_A)(v))$ and $\ell'_B(v) := RELABEL(\ell_B, f(V_B, E_B)(v))$ cannot be distinguished by the 1-WL algorithm.*

*Proof.* The graphs $A$ and $B$ are the graphs in Fig. 1c and Fig. 1d. Note that without the colors, both graphs are isomorphic, and all nodes are in one orbit, which is verified by the isomorphism in cycle notation $(1, 2, 3, 4, 5, 6, 7, 8, 9, 10, 11, 12)$. This means that $\ell_A(v) = \ell_A(w)$ if and only if $\ell'_A(v) = \ell'_A(w)$ and $\ell_B(v) = \ell_B(w)$ if and only if $\ell'_B(v) = \ell'_B(w)$. Which implies that regardless of the specific function $f$, the partition of colors from $\ell_A$ and $\ell_B$ to $\ell'_A$ and $\ell'_B$ does not change and thus Fig. 1c and Fig. 1d also represent graphs $A'$ and $B'$. Lastly, note that all nodes in Fig. 1c and Fig. 1d have two black and two red neighbors, which implies that these colors already represent the stable color partition $C^1_\infty$ and thus 1-WL cannot distinguish these graphs. □

Note that, Theorem 2 applies to all PSEs that do not involve a random variable as output. Further, this means that if GPSE (assuming no randomness in its predictions) or other PSEs are used to process graphs that are more than 1-WL hard to distinguish, then the downstream GNN should be chosen based on expressivity, for instance, by also adding RNF into its input.

**GPSE⁻.** While the GPSE⁺variant considers strengthening the pre-training stage with hard graphs, our experimental results in Section 5 on real-world datasets do not read improved performance. Therefore, to address the question of the importance of expressivity in GPSE, we also replace the input RNF with a constant feature vector, effectively removing the universality offered by RNF. We call this variant GPSE⁻. It has the same model architecture as GPSE; however, instead of random normally distributed inputs, it uses $\mathbf{1}$ vectors, i.e., $\text{GPSE}^-(G)(w) := \text{GPSE}(G, \mathbf{1})(w)$. In practice, this implies that the GPSE⁻model is not capable of computing all PSEs for any input graph. As an example, GPSE⁻cannot compute cycle counts for the graphs in Fig. 1a and Fig. 1b, since GPSE⁻cannot tell these two graphs apart. However, since molecules are almost trees (Ahn et al., 2022), GPSE⁻should be able to still learn the target PSEs for the pre-training dataset, MolPCBA, as well as for the other evaluation datasets like ZINC-12k. Notably, Table 1 shows that GPSE⁻does not excel on datasets that involve graphs harder than 1-WL. In particular, the performance of GPSE⁻is roughly in line with a constant model. On TRIX GPSE⁻outperforms GPSE as well as GPSE⁺, because these other variants are having trouble generalizing well to the larger 3-regular graphs of TRIX, and actually perform worse than a naive, constant predictor. The evaluations in Section 5 will demonstrate that, indeed, RNF is not necessary for good practical performance either. In practice, at least for current datasets, GPSE, GPSE⁺, and GPSE⁻perform comparably.

**Practical implications.** As will be demonstrated in the next section, the differences between GPSE, GPSE⁺, and GPSE⁻in practice are virtually non-existent. This is likely due to the fact that expressivity is rarely relevant in practice. However, if expressivity is determined to be irrelevant GPSE⁻should probably be preferred, as the use of RNF in GPSE could lead to unpredictable results. While on the other hand,

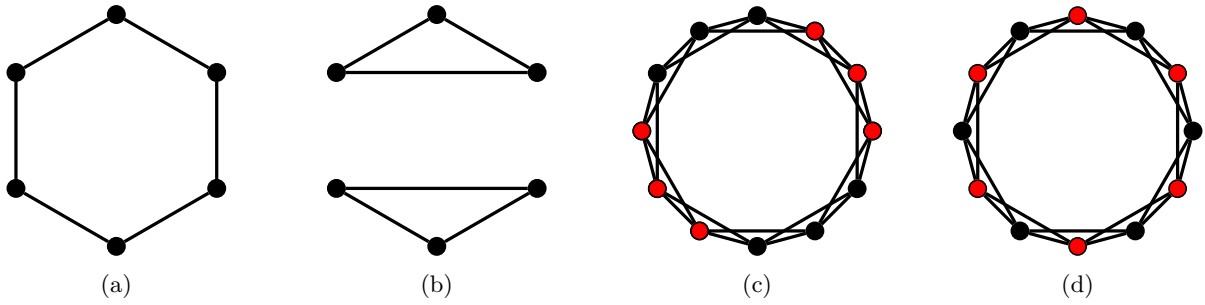

(a)          (b)          (c)          (d)

Figure 1: GPSE without RNF cannot differentiate graphs (a) and (b). An MPNN with the orbit partition as node features cannot differentiate graphs (c) and (d).

Table 2: Using GPSE on ZINC-12k significantly reduces the number of epochs needed to reach the same test performance as other PSEs. This table shows the factor of speedup (in terms of epochs) of GPSE to reach the same MAE on ZINC-12k on various downstream GNNs and PSEs. For instance, 2 means GPSE reaches the same performance with only half of the epochs. This table comparing GPSE$^-$and GPSE$^+$to other PSEs can be found in Appendix G.

| PSE↓ \ Downstream→ | GCN | GatedGCN | GIN | GPS |
|---|---|---|---|---|
| none | 71.2±10.9 | 65.6±12.2 | 61.0±10.2 | 12.4±1.4 |
| rand | 905.0±0.0 | 110.0±0.0 | 253.0±0.0 | 98.5±14.1 |
| LapPE | 43.8±4.4 | 45.5±6.5 | 31.9±4.7 | 14.7±1.7 |
| RWSE | 20.5±2.4 | 25.1±3.1 | 21.5±3.4 | 2.9±0.5 |
| AllPSE | 13.5±2.3 | 20.1±3.4 | 18.7±2.3 | 2.6±0.2 |

if expressivity is important, then an approach closer to GPSE$^+$should be taken, specifically focusing on pretraining using hard graphs that are relevant to the task or generally hard graphs.

## 5 Experimental Results

We evaluate PSEs as well as the proposed GPSE variants from Section 4 across three critical aspects of performance. First, we analyze the speed at which GPSE-augmented GNNs converge to the performance of other PSE-augmented GNNs. Specifically, we calculate the number of epochs required by GPSE to reach the performance attained by other baseline GNNs. Second, we explore the transferability of PSEs in data-scarce environments. To this end, we plot the amount of data (i.e., sample size) vs. performance, investigating how well PSE-augmented models perform with varying amounts of training data. Finally, we investigate why GPSE improves performance over other PSEs for most but not all datasets. We focus on ZINC-12k, the perhaps clearest example of good GPSE performance, and consider generalization by inspecting the gap between training and test metrics.

Overall, our experiments include three dataset types: (i) synthetic datasets EXP, CEXP, CSL, TRI, and TRIX (Abboud et al., 2021; Murphy et al., 2019; Sato et al., 2021), designated to study the expressivity of GPSE, as reported in Table 1. (ii) ZINC-12k, where we find GPSE to perform well across all considered aspects, and (iii) MolNet, where we find that GPSE does not perform better than baseline methods in general. The evaluations on these diverse datasets allow us to provide well-grounded insights regarding the utilization of GPSE as a component in future GFMs. We provide details on the learning and evaluation procedures in Appendix E. We elaborate on the considered PSEs in Appendix D. We elaborate on the datasets and the motivation for using them in Appendix F.

### 5.1 GPSEs Convergence to PSEs Performance

The efficiency of neural network optimization is critical, particularly in minimizing the time required for training models. Reducing the training time can result in significant resource savings.

To evaluate the speedup GPSE enables, we compare how many epochs a GPSE-augmented GNN takes to reach the same performance as other PSE-augmented GNNs, as shown in Table 2. Our results show that using GPSE yields good performance significantly faster compared with all other PSEs. The smallest speedup is a factor of 2, and in almost all cases, GPSE obtains a speedup factor of 10 or more. Reaching good performance faster not only saves training time during parameter tuning but also reduces the environmental impact by lowering $CO_2$ emissions associated with prolonged computational efforts. We note here, that Table 2 does not imply that GPSE converges to its optimal performance faster, it only implies that GPSE attains good performances faster than other PSEs. In fact, according to our evaluations, GPSE converges at the same speed as other PSEs (see Appendix G).

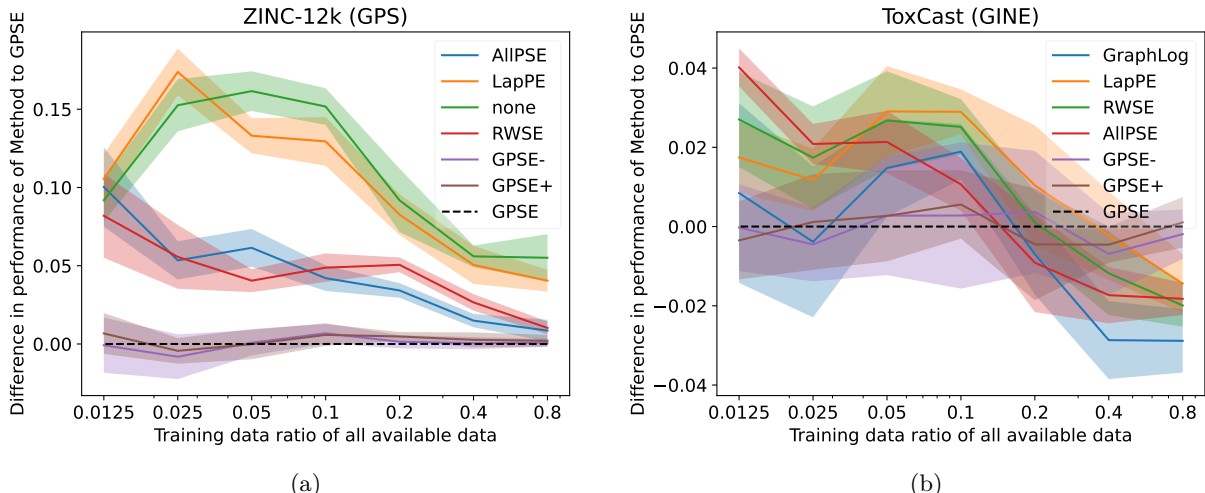

Figure 2: GPSE variants outperform all other PSEs regardless of the amount of available training data on ZINC-12k. Notably, for less available training data, the advantage of GPSE and its variants is more strongly pronounced. However, for ToxCast the opposite is true. This figure shows downstream training with fractions of the ZINC-12k and ToxCast datasets. We show the difference in performance (MAE ↓ for ZINC-12K and AUROC ↑ for ToxCast) obtained by various PSEs, including GPSE$^-$and GPSE$^+$. We show results on additional datasets in Fig. 3.

## 5.2 Influence of Sample Size

A common challenge in machine learning is the scarcity of available training data, which can significantly hinder model performance. It is common in areas like NLP and CV to finetune pre-trained models in data-scarce applications. Some prominent examples are LLMs like GPT-3, which can be fine-tuned or used in zero-shot settings for various NLP tasks (Brown et al., 2020). In CV, models like CLIP, which learn visual concepts from natural language supervision, have been effectively adapted to new tasks with minimal data (Radford et al., 2021). These approaches leverage powerful representations learned from large-scale data, making them highly effective in transfer learning scenarios.

To investigate how pre-trained GPSE models can mitigate the impact of limited data, we conducted a few-shot learning-inspired experiment. In this experiment, we progressively reduce the amount of training data for the downstream task, and evaluate the resulting performance of a downstream GNN when combined with different PSEs against GPSE and its variants. As proposed in an ablation study in Cantürk et al. (2024), we start with all training data (80% training, 10% validation, 10% testing) and iteratively halve the training data to just 1.25%, while leaving the validation and test data unchanged.

Fig. 2 shows the results of the experiment on ZINC-12k and MolNet ToxCast datasets. GPSE's performance is used as a baseline (black dotted line) and compared with other PSEs, including LapPE, RWSE, and the GPSE$^-$and GPSE$^+$variants. Both GPSE$^-$and GPSE$^+$perform similarly to GPSE. On ZINC-12k, GPSE-like models outperform all other PSEs, with the gap widening as data decreases logarithmically. However, this advantage is not consistent across all datasets. For instance, on ToxCast, GPSE-like models outperform other PSEs with full training data but are surpassed by other PSEs in data-scarce scenarios. Additional results are provided in Fig. 3 (Appendix H), highlighting the significant benefits of learned PSEs in future GFMs.

## 5.3 Generalization of PSEs

We now investigate why GPSE might or might not outperform other PSEs when used to augment downstream GNN models. We focus on two possible proxies for good performance: (i) better training loss and (ii) better generalization error. Notably, if one is kept constant, then the respective other will govern changes in

Table 3: GPSE variants offer the largest (best) generalization error. This table shows the generalization error (train − test MAE) on ZINC-12k for various PSEs and downstream MPNN choices. The largest (best) values are **bold**, and insignificantly smaller values are underlined.

| PSE ↓ \ Downstream→ | GCN | GatedGCN | GIN | GPS |
|---|---|---|---|---|
| none | -0.143±0.010 | -0.137±0.013 | -0.131±0.011 | -0.085±0.009 |
| rand | -0.136±0.045 | -0.234±0.041 | -0.152±0.021 | -0.158±0.050 |
| LapPE | -0.112±0.008 | -0.118±0.009 | -0.110±0.007 | -0.065±0.032 |
| RWSE | -0.091±0.007 | -0.088±0.010 | -0.073±0.013 | -0.048±0.006 |
| AllPSE | -0.075±0.006 | -0.078±0.009 | -0.060±0.010 | -0.050±0.007 |
| GPSE | **-0.032±0.005** | -0.045±0.004 | -0.034±0.004 | **-0.037±0.007** |
| GPSE$^-$ | -0.033±0.005 | -0.042±0.005 | **-0.034±0.005** | -0.042±0.005 |
| GPSE$^+$ | -0.034±0.006 | **-0.040±0.005** | -0.035±0.004 | -0.041±0.005 |

Table 4: GPSE does not always offer the lowest (best) generalization error. This table shows generalization error (train − test AUROC) on MolNet datasets. Smallest (best) values are **bold**, and insignificantly larger values are underlined.

| PSE ↓ \ Dataset→ | BACE | BBBP | ClinTox | HIV | MUV | SIDER | Tox21 | ToxCast |
|---|---|---|---|---|---|---|---|---|
| GraphLog | 17.3±1.6 | 32.0±1.2 | 21.7±3.6 | 16.3±5.7 | 15.9±3.7 | 12.2±3.8 | 20.5±3.2 | **17.1±2.7** |
| LapPE | **6.1±5.0** | 27.6±2.1 | 18.0±4.5 | 13.1±6.3 | 14.6±3.1 | 11.8±1.5 | 14.1±1.2 | 17.4±1.4 |
| RWSE | 11.3±2.2 | 28.4±2.4 | 17.3±2.4 | 12.9±1.8 | 10.4±1.6 | 12.7±2.5 | 18.4±2.2 | 18.5±1.8 |
| AllPSE | 13.3±2.7 | **23.3±3.4** | 21.3±2.9 | 19.8±4.8 | **10.0±1.7** | 13.2±0.9 | **12.4±1.6** | 19.3±1.9 |
| GPSE | 10.6±2.4 | 25.6±4.3 | **16.8±7.4** | 17.2±4.8 | 17.6±0.7 | 12.4±3.0 | 13.5±1.8 | 19.1±1.3 |
| GPSE$^-$ | 9.9±3.6 | 26.3±3.8 | 17.4±5.1 | 13.8±6.7 | 17.9±2.3 | 11.5±2.7 | 12.7±2.1 | 18.4±2.2 |
| GPSE$^+$ | 12.3±4.0 | 26.5±3.2 | 18.2±4.9 | **12.8±4.7** | 16.9±2.0 | **9.3±3.8** | 14.4±1.9 | 19.6±0.7 |

test performance. In this set of experiments, we define the generalization error as the difference between train performance and test performance, where the performance metric depends on the dataset and task of interest. For example, in the case of regression tasks such as on ZINC-12k, a larger value—based on MAE—means better generalization, while for classification tasks, a lower value—based on AUROC–indicates better generalization. We provide the obtained generalization errors for ZINC-12k in Table 3, and the respective train (and test) mean-absolute-error (MAE) in Table 5 (and Table 16 in Appendix I). On ZINC-12k the GPSE variants attain the best test performance regardless of the backbone (Table 16 in Appendix). Table 3 shows the GPSE variants attain the best generalization errors compared with other PSEs. In contrast, Table 5 shows that while GPSE attains fairly low training MAE, other PSEs "outperform" the GPSE variants in terms of training performance. These observations imply that the performance gains of GPSE-augmented models are primarily due to better generalization, and not, for instance, due to a model that better fits the training set. However, this better generalization does not always hold. In contrast to previous observations, Table 4 showing the generalization error for the MolNet datasets, demonstrates that GPSE and its variants can be outperformed considering generalization on some downstream tasks (MUV and BACE are notable examples). Better generalization and generally better performance are desired properties of a foundation model. These results indicate that while GPSE shows some superior generalization capabilities, they need to be examined per case. While this work studied expressivity in Section 4 and generalization here, we are not attempting to conflate the two herein. The observations on expressivity and generalization are entirely separate. Notably, even though GPSE$^-$, GPSE, and GPSE$^+$ have different practical expressivity, their generalization seems comparable, which agrees with previous work on the complexity of expressivity and generalization (Franks et al., 2024).

Table 5: GPSE variants do not offer the smallest (best) training error. This table shows the MAE on the trainset, which was also used as loss, on ZINC-12k for various PSEs and downstream MPNN choices. Smallest (best) values are **bold**, and insignificantly larger values are underlined.

| PSE ↓ \ Downstream→ | GCN | GatedGCN | GIN | GPS |
|---|---|---|---|---|
| none | 0.146±0.012 | 0.107±0.017 | 0.148±0.014 | 0.038±0.012 |
| rand | 1.119±0.067 | 0.988±0.049 | 1.090±0.021 | 0.711±0.054 |
| LapPE | 0.106±0.010 | 0.071±0.009 | 0.107±0.009 | 0.076±0.107 |
| RWSE | 0.086±0.008 | 0.078±0.011 | 0.102±0.011 | 0.027±0.008 |
| AllPSE | **0.077±0.006** | **0.064±0.009** | **0.089±0.010** | **0.021±0.006** |
| GPSE | 0.093±0.005 | 0.067±0.005 | 0.093±0.004 | 0.033±0.010 |
| GPSE$^-$ | 0.096±0.006 | 0.071±0.007 | 0.096±0.003 | 0.024±0.006 |
| GPSE$^+$ | 0.093±0.007 | 0.073±0.004 | 0.092±0.004 | 0.025±0.006 |

## 6 Discussion and Conclusion

This paper explored PSEs on graphs, particularly the recently proposed GPSE. We found that the primary factor influencing PSE performance is their generalization ability, highlighting the need for a deeper understanding of strong generalization in GNNs—a critical property for GFMs. Although GPSE offers a promising approach, it does not consistently generalize well across tasks and graph structures, underscoring the need for models with improved generalization.

Future research should enhance generalization in GNNs, especially for graph SEs and PEs. Initial studies on the generalization of PSEs and embeddings for substructures, such as in Franks et al. (2024), suggest that subgraph-enhanced WL kernels' generalization depends on the counted substructures. This may apply to MPNNs as well; for instance, PSEs used in GPSE pre-training generalize well on ZINC but poorly on some MolNet datasets. The broader impact of graph structure on generalization has been identified as a key challenge in ML on graphs (Challenge 3.2 in Morris et al. (2024)), and addressing these challenges will be crucial for advancing GFMs.

## Acknowledgements

Part of this work was conducted within the DFG Research Unit FOR 5359 on Deep Learning on Sparse Chemical Process Data (BU 4042/2-1, KL 2698/6-1, and KL 2698/7-1) and the DFG TRR 375 (no. 511263698). SF and MK acknowledge support by the Carl-Zeiss Foundation (AI-Care, Process Engineering 4.0), the BMBF award 01|S2407A, and the DFG awards FE 2282/6-1, FE 2282/1-2, KL 2698/5-2, and KL 2698/11-1. ME is funded by the Blavatnik-Cambridge fellowship, the Cambridge Accelerate Programme for Scientific Discovery and the Maths4DL EPSRC Programme. Additional support is by Bourse en intelligence artificielle des Études supérieures et postdoctorales (ESP) 2023-2024 [SC]; Canada CIFAR AI Chair, IVADO (Institut de valorisation des données) grant PRF-2019-3583139727, FRQNT (Fonds de recherche du Québec - Nature et technologies) grant 299376, NSERC Discovery grant 03267 and NSF DMS grant 2327211 [GW].

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

## A  Expressivity

For the most part, in this work, we only consider expressivity attained from RNF because it was proposed as the method of attaining expressivity in Cantürk et al. (2024). Here we now discuss the reason for this choice as well as some related work on the topic of expressivity in machine learning on graphs. Related work on expressivity has focused on two main directions for attaining expressivity for MPNNs.

The first direction, which we discussed in detail in this paper, is essentially the use of PSEs. Notably PSEs like LapPE (Kreuzer et al., 2021; Rampášek et al., 2022) and RWSE (Rampášek et al., 2022; Dwivedi and Bresson, 2021; Dwivedi et al., 2022b) increase expressivity when used to augment downstream MPNNs but have not been shown to be universal and in fact we show in this work that they cannot be universal. Further PSEs that are well-known to be universal are based on subgraph counting (Shiv and Quirk, 2019; Bouritsas et al., 2022; Zhao et al., 2022; Chen et al., 2022) or homomorphism counting (Jin et al., 2024; Bouritsas et al., 2022; Zhang et al., 2024). However, related work usually considers the setting of uncolored graphs, in which case these PSE are indeed universal if sufficiently many subgraphs/homomorphisms are being counted. Once more, according to Theorem 2 subgraph and homomorphism counting that disregards node colors will not grant universality to augmented downstream MPNNs considering colored graphs. The emphasis here is on PSEs that disregard node colors, one could propose subgraph and homomorphism counting of colored subgraphs which would be fully universal. Lastly, RNF which can be considered a PSE has been shown to grant universality when used to augment downstream MPNNs (Abboud et al., 2021; Sato et al., 2021; Franks et al., 2023). As RNF relies on randomness to differentiate nodes during computation, RNF is universal regardless of node colors. The universality or expressivity of PSEs is inherently linked to the individualization refinement algorithm for which PSEs act as individualization functions as discussed in Franks et al. (2023), which is inherently efficient assuming that any precomputations of the PSEs can be done efficiently.

The second direction modifies an MPNN's computation to attain more expressivity. Common examples include the k-GNN (Morris et al., 2019; 2020; Wang et al., 2022b) and PPGN (Maron et al., 2019a;b). These higher-order methods are essentially related to the $k$-dimensional Weisfeiler-Leman algorithm and, as a result, suffer from high computational and memory costs. However, they notably show advantages in performance in some cases.

GPSE (Cantürk et al., 2024) used RNF instead of other methods of expressivity. Notably, of the methods mentioned, only RNF, subgraph, and homomorphism counting, and higher-order methods are truly universal. Subgraph and homomorphism counting, however, require a potentially very large set of graphs to be counted (which is even bigger if not infinitely large for colored graphs), and higher-order methods might require a very large order (in the case of k-GNNs, a very large $k$) for which the computational and memory cost can be prohibitively large. In this sense, then, RNF is the only simple, effective, and scalable method of granting GPSE universality, which makes it the best choice for GPSE.

Lastly, in this work we focus mostly on expressivity as it pertains to the 1-WL, however, some more fine-grained approaches to expressivity have been recently proposed. For example, expressivity based on biconnectivity (Zhang et al., 2023) or expressivity based on homomorphism counting (Zhang et al., 2024).

## B  Proof of Theorem 1

**Theorem 1.** *Under mild assumptions on the input graphs, an MPNN consisting of sufficiently many GatedGCN layers can approximate an MPNN made up of GIN layers arbitrarily well.*

*Proof.* A GIN layer is defined as follows:

$$\boldsymbol{h}_u^{(l+1)} = \mathrm{MLP}\left((1+\epsilon)\boldsymbol{h}_u^{(l)} + \sum_{\{u,v\}\in E(G)} \boldsymbol{h}_v^{(l)}\right).$$

For simplicity we will assume the MLP to be made up of two linear transformations with ReLU activation. However, the following proof can also be modified for other MLP choices.

For the GatedGCN layers, we will assume that the input graphs node features contain one feature that is constant for all nodes, i.e., if the input graph is $G = (V, E)$ with node features $X : V \to \mathbb{R}^d$, without loss of generally, $\forall v \in V : X(v)_0 = 1$. We will ensure that this is also true for the output after multiple GatedGCN layers. This means that we will be working with the invariance that $\forall v, l' : (h_v^{(l')})_0 = 1$. Now, note that one GatedGCN layer is defined as

$$h_u^{(l+1)} = \text{ReLU}(U^{(l)} h_u^{(l)} + \sum_{\{u,v\} \in E(G)} \eta_{uv}^{(l)} \odot V^{(l)} h_v^{(l)})$$

with

$$\eta_{uv}^{(l)} := \sigma(A^{(l)} h_u^{(l)} + B^{(l)} h_v^{(l)}).$$

We first define a GatedGCN layer that approximates $(1 + \epsilon) h_u^{(l)} + \sum_{\{u,v\} \in E(G)} h_v^{(l)}$. Then, two GatedGCN layers are defined that compute the MLP. For this assume that we are trying to approximate layer $l$ of the GIN and are defining layer $l'$ of the GatedGCN. By abuse of notation, we will refer to the hidden representations of the GIN of node $i$ in layer $l$ as $h_u^{(l)}$ and refer the same for the GatedGCN as $h_u^{(l')}$. Note that, if we are approximating $h_u^{(l)}$ by multiple GatedGCN layers starting at $h_u^{(l')}$, then $h_u^{(l')} \approx \begin{bmatrix} 1 \\ h_u^{(l)} \end{bmatrix}$, with equality in the first component.

Let $1 > \alpha > 0$, then by choosing $A := \begin{bmatrix} \sigma^{-1}(1 - \alpha)\mathbf{1} & 0 \end{bmatrix}$, that is the first column is filled with $\sigma^{-1}(1 - \alpha)$ and it is 0 everywhere else, and $B := 0$, $\sigma(A^{(l')} h_u^{(l')} + B^{(l')} h_v^{(l')}) = (1 - \alpha)\mathbf{1}$. Let $\beta := (1 + \epsilon)(1 - \alpha)$, then choosing $V^{(l')} = \begin{bmatrix} 0 & \mathbf{0}^T \\ \mathbf{0} & I \\ \mathbf{0} & -I \end{bmatrix}$ and $U^{(l')} = \begin{bmatrix} 1 & \mathbf{0}^T \\ \mathbf{0} & \beta I \\ \mathbf{0} & -\beta I \end{bmatrix}$,

$$\text{ReLU}(U^{(l')} h_u^{(l')} + \sum_{\{u,v\} \in E(G)} \eta_{ij}^{(l')} \odot V^{(l')} h_v^{(l')}) = \begin{bmatrix} 1 \\ \text{ReLU}((1 - \alpha) h_u^{(l)}) \\ \text{ReLU}(-(1 - \alpha) h_u^{(l)}) \end{bmatrix}.$$

Notice that

$$\begin{bmatrix} \mathbf{0} & I & -I \end{bmatrix} \text{ReLU}(U^{(l')} h_u^{(l')} + \sum_{\{u,v\} \in E(G)} \eta_{ij}^{(l')} \odot V^{(l')} h_v^{(l')})$$

$$= (1 - \alpha) \left( (1 + \epsilon) h_u^{(l)} + \sum_{\{u,v\} \in E(G)} h_v^{(l)} \right).$$

From here on refer to

$$p_i := (1 + \epsilon) h_u^{(l)} + \sum_{\{u,v\} \in E(G)} h_v^{(l)}$$

and

$$p_i' := \text{ReLU}(U^{(l')} h_u^{(l')} + \sum_{\{u,v\} \in E(G)} \eta_{ij}^{(l')} \odot V^{(l')} h_v^{(l')}).$$

Now assume that the MLP of the GIN in layer $l$ is defined as follows:

$$\text{MLP}(x) := \text{ReLU}(W_2 \text{ReLU}(W_1 x)).$$

We can choose $V^{(l'+1)} = 0$, a zero matrix, and $U^{(l'+1)} = \begin{bmatrix} 1 & \mathbf{0}^T & \mathbf{0}^T \\ \mathbf{0} & W_1 & -W_1 \end{bmatrix}$, then regardless of the choice of $A^{(l'+1)}$ and $B^{(l'+1)}$,

$$\text{ReLU}(U^{(l'+1)}p_i' + \sum_{\{u,v\}\in E(G)} \eta_{ij}^{(l'+1)} \odot V^{(l'+1)}p_j') = \text{ReLU}(U^{(l'+1)}p_i')$$

$$= \begin{bmatrix} 1 \\ \text{ReLU}((1-\alpha)W_1 p_i) \end{bmatrix}$$

$$= \begin{bmatrix} 1 \\ (1-\alpha)\text{ReLU}(W_1 p_i) \end{bmatrix} =: p_i''.$$

Similarly, choosing $V^{(l'+2)} = 0$ and $U^{(l'+2)} = \begin{bmatrix} 1 & \mathbf{0}^T \\ \mathbf{0} & W_2 \end{bmatrix}$,

$$\text{ReLU}(U^{(l'+2)}p_i'' + \sum_{\{u,v\}\in E(G)} \eta_{ij}^{(l'+2)} \odot V^{(l'+2)}p_i'') = \begin{bmatrix} 1 \\ \text{ReLU}(W_2(1-\alpha)\text{ReLU}(W_1 p_i)) \end{bmatrix}$$

$$= \begin{bmatrix} 1 \\ (1-\alpha)\text{ReLU}(W_2\text{ReLU}(W_1 p_i)) \end{bmatrix}$$

$$= \begin{bmatrix} 1 \\ (1-\alpha)h_i^{(l+1)} \end{bmatrix}.$$

Considering the difference between the relevant components, we get that the approximation error is

$$\max_{l,i}\|h_i^{(l)} - h_i^{(l')}\| = \max_{l,i}\|h_i^{(l)} - (1-\alpha))h_i^{(l)}\|$$

$$= \max_{l,i}\|\alpha h_i^{(l)}\|$$

$$= \alpha \max_{l,i}\|h_i^{(l)}\|.$$

Assuming that $\max_{l,i}\|h_i^{(l)}\|$ is bounded, this quantity can be arbitrarily small by choosing $\alpha$ small enough. To be clear, the assumption on the input we require is that $\max_{l,i}\|h_i^{(l)}\|$ is bounded and that each node carries a feature that is constant over all nodes, in this proof we assumed this constant to be 1, however, this is true for any such non-zero constant. Notably, we can also make weaker assumptions, by assuming that there is a linear combination of features that is constant over all nodes. However, in practice, we can simply add a constant 1 as a feature to each node, which is a common augmentation to ensure MPNNs can count the size of neighborhoods. □

## C  Additional expressivity experiment

Here we include an additional expressivity experiment based on Huang et al. (2023b). The task is to count the inclusion of nodes in certain substructures on random Erdős-Rényi graphs. As in the original work, we train 5-layer GatedGNNs (Li et al., 2013). We run the experiment with 10 different random seeds and report the test MAE divided by the label standard deviation referred to as normalized MAE in Table 6 and Table 7. Since GPSE is trained to predict cycle counts, we would expect GPSE to perform well in Table 6. However, as demonstrated previously, GPSE only learns to count cycles for molecule-like graphs, because these are the only graphs included in the pre-training set. GPSE$^+$has additional hard graphs added to its pre-training set, which appears to help its performance somewhat, although its performance is still a far cry from other methods reported in Huang et al. (2023b). Perhaps adding Erdős-Rényi graphs instead of random regular graphs would help with its performance. In some instances GPSE$^-$performs best, which might be due to GPSE$^-$overfitting less to the use of RNI.

Table 6: The inclusion of hard graphs in GPSE$^+$pre-training is crucial to the model's success on expressivity datasets. GPSE$^+$performs noticeably better for 3-cycle and 4-cycle counting, while for 5-cycle and 6-cycle counting, the differences between the methods are reduced. Notably, GPSE$^-$outperforms GPSE, indicating that perhaps the use of RNI without adequate pre-training graphs leads to overfitting. We report the normalized mean absolute error ↓ of GPSE$^-$, GPSE, and GPSE$^+$of counting different subgraphs. This experiment is based on Huang et al. (2023b).

| Method↓  \ Task→ | 3-cycle | 4-cycle | 5-cycle | 6-cycle |
|---|---|---|---|---|
| GPSE- | 0.208±0.002 | 0.167±0.001 | 0.155±0.001 | **0.117±0.001** |
| GPSE | 0.220±0.002 | 0.174±0.002 | 0.161±0.001 | 0.122±0.001 |
| GPSE+ | **0.147±0.002** | **0.156±0.001** | **0.152±0.001** | 0.122±0.001 |

Table 7: The inclusion of hard graphs in GPSE$^+$pre-training is crucial to the model's success on expressivity datasets. GPSE$^+$performs noticeably better for TT and CC counting while the differences between the methods for 4-clique and 4-path counting reduce. GPSE$^-$performs noticeably best for TR. We report the normalized mean absolute error ↓ of GPSE$^-$, GPSE, and GPSE$^+$of counting different subgraphs. TT refers to a tailed triangle. CC refers to a chordal cycle. TR refers to a triangle rectangle. This experiment is based on Huang et al. (2023b).

| Method↓  \ Task→ | TT | CC | 4-Clique | 4-Path | TR |
|---|---|---|---|---|---|
| GPSE- | 0.244±0.002 | 0.195±0.002 | **0.147±0.001** | **0.063±0.000** | **0.160±0.002** |
| GPSE | 0.260±0.001 | 0.208±0.002 | 0.148±0.001 | 0.065±0.001 | 0.174±0.002 |
| GPSE+ | **0.200±0.001** | **0.182±0.002** | 0.149±0.001 | 0.067±0.000 | 0.168±0.001 |

## D  PSEs

We consider a simple undirected and unweighted graph $G = (V, E)$ with vertex set $V$ and edge set $E$, and no node or edge features. The number of nodes and edges are denoted with $n = |V|$ and $m = |E|$, respectively. The corresponding adjacency matrix of graph $G$ is $\mathbf{M} \in \{0, 1\}^{n \times n}$, where $\mathbf{M}_{ij} = 1$ if $\{v_i, v_j\} \in E$ and 0 otherwise. The graph Laplacian $\mathbf{L}$ is then defined as

$$\mathbf{L} = \mathbf{D} - \mathbf{M} \tag{7}$$

where $\mathbf{D} \in \mathbb{N}^{n \times n}$ is the diagonal degree matrix such that $\mathbf{D}_{ii} = deg(v_i) = |\mathcal{N}(v_i)| = |\{u|(v_i, u) \in E\}|$.

The graph Laplacian is a real symmetric matrix, with its full eigendecomposition as

$$\mathbf{L} = \mathbf{U}\boldsymbol{\Lambda}\mathbf{U}^\top \tag{8}$$

where, $\boldsymbol{\Lambda}_{ii} = \lambda_i$ and $\mathbf{U}_{[:,i]} = u_i$ are the $i^{\text{th}}$ eigenvalue and eigenvector (an eigenpair) of the graph Laplacian. We sort the eigenpairs from the smallest to largest eigenvalue, i.e., $0 = \lambda_1 \leq \lambda_2 \leq \cdots \leq \lambda_n$. We further denote $\hat{\mathbf{U}}$ (and analogously the subdiagonal matrix $\hat{\boldsymbol{\Lambda}}$) as the matrix of Laplacian eigenvectors corresponding to non-trivial eigenvalues.

$$\hat{\mathbf{U}} = \mathbf{U}_{[:,\{i|\lambda_i \neq 0\}]} \tag{9}$$

Finally, we denote the ($\ell_2$) normalization operation as $\text{normalize}(x) := \frac{x}{\|x\|_2}$

In Cantürk et al. (2024), the node-level PSEs presented below are normalized to zero mean and unit standard deviation per graph when training GPSE; we follow this for training stability in training GPSE$^+$and GPSE$^-$. When used as standalone or combined benchmarks, the PSEs are not normalized.

**Laplacian eigenvector positional encodings (LapPE)**

LapPE is defined as the absolute value of the $\ell_2$ normalized eigenvectors associated with non-trivial eigenvalues. By default, we use the first four LapPE to train GPSE.

$$\text{LapPE}_i = |\,\text{normalize}(\hat{\mathbf{U}}_{[:,i]})|\qquad(10)$$

The absolute value operation is needed to counter the sign ambiguity of the graph Laplacian eigenvectors, a known issue to many previous works that use the Laplacian eigenvectors to augment the models (Dwivedi et al., 2020; Lim et al., 2023). We maintain learning this absolute value function of the eigenvectors as per Cantürk et al. (2024), and similarly learn the eigenvalues as graph-level attributes.

**Random walk structural encodings (RWSE)**  The random walk matrix is defined as the row-normalized adjacency matrix $\mathbf{P} := \mathbf{D}^{-1}\mathbf{M}$. $\mathbf{P}_{i,j}$ corresponds to the transition probability from $v_i$ to $v_j$ at a given step.

The $k^{\text{th}}$ RWSE (Dwivedi et al., 2022b) is defined as the probability of returning back to the starting state of a random walk after $k$ steps of a random walk:

$$\text{RWSE}_k = diag(\mathbf{P}^k)\qquad(11)$$

**AllPSE**  Cantürk et al. (2024) compare the learned encodings (GPSE) with combinations of multiple PSEs in addition to the standalone LapPE and RWSE for a fairer comparison. Two "combined" encodings are considered: Combining the two standalone encodings as LapPE+RWSE, and AllPSE, which represents a combination of *all* PSEs used in GPSE training. The following encodings are thus not used in standalone form, but are relevant in that they are essential to GPSE training as well as the AllPSE benchmark.

**Electrostatic potential positional encodings (ElstaticPE)**  ElstaticPE treats each node as a charged particle to compute electrostatic potentials between node pairs, based on the pseudoinverse of the graph Laplacian $\mathbf{L}^{\dagger}$ with the diagonal set to 0 (so that the potential of each node each node's potential on itself is 0) resulting in:

$$\mathbf{Q} = \mathbf{L}^{\dagger} - diag(\mathbf{L}^{\dagger})\mathbf{1}_n\qquad(12)$$

ElstaticPE for any node $v_i$ is then a collection of statistics that summarizes the electrostatic interaction of $v_i$ with other nodes, such as the minimum, mean and standard deviations of potentials from $v_i$ to all other nodes $v_j$ or only its neighbors $v_k \in \mathcal{N}(v_i)$ (Kreuzer et al., 2021). We defer to Cantürk et al. (2024) for more detailed definitions of its individual components.

**Heat kernel diagonal structural encodings (HKdiagSE)**  As defined in Cantürk et al. (2024):

$$\text{HKdiagSE}_k = \sum_{i:\lambda_i \neq 0} e^{-k\lambda_i}\,\text{normalize}(\mathbf{U}_{[:,i]})^2\qquad(13)$$

**Cycle counting structural encodings (CycleSE)**  CycleSE counts the number of $k$-cycles in the graph (e.g., 3-cycles correspond to triangles in the graph) to encode global graph structure. Similar to LapPE eigen*values*, CycleSE is learned as a graph-level regression task in GPSE training.

$$\text{CycleSE}_k = |\{\text{Cycles of length k}\}|\qquad(14)$$

# E  Training and Evaluation Details

This work is built upon Cantürk et al. (2024) and uses the exact same experimental setup as well as largely the same code. The code itself is modified to accommodate the additional experiments proposed here. That is, for the experiments shown in Fig. 2 and Fig. 3, as well as the additional datasets CEXP, TRI, and TRIX, presented in Table 1. We thus refer to Cantürk et al. (2024) for further details beyond those presented here. Details on model design are shown in Table 8 and Table 9.

Table 8: GPS+method hyperparameters for molecular property prediction benchmarks

| Hyperparameter | ZINC-12k |
|---|---|
| # GPS Layers | 10 |
| Hidden dim | 64 |
| GPS-MPNN | GINE |
| GPS-SelfAttn | – |
| # Heads | 4 |
| Dropout | 0.00 |
| Attention dropout | 0.50 |
| Graph pooling | mean |
| PE dim | 32 |
| PE encoder | 2-Layer MLP |
| Input dropout | 0.50 |
| Output dropout | 0.00 |
| Batchnorm | yes |
| Batch size | 32 |
| Learning rate | 0.001 |
| # Epochs | 200 |
| # Warmup epochs | 50 |
| Weight decay | 1.00e-5 |
| # Parameters | 292,513 |
| PE precompute | 2 min |
| Time (epoch/total) | 10s/5.78h |

**MoleculeNet small benchmarks settings** We used the default GINE architecture following previous studies (Hu* et al., 2020), which has five hidden layers and 300 hidden dimensions. For all five benchmarks, we use the same GPSE processing encoder settings as shown in Table 9a.

**CSL, EXP, and TRI synthetic graph benchmarks settings** We follow He et al. (2023) and use GIN (Xu et al., 2019) as the underlying MPNN model, with five hidden layers and 128 dimensions. We use the same GPSE processing encoder settings for CSL, EXP, CEXP, TRI, and TRIX as shown in Table 9b.

## E.1  Convergence to PSEs Performance

This experiment is modeled after the experiment in Table 3 of Cantürk et al. (2024). Different downstream models (GCN, GatedGCN, GIN, and GPS) are trained using varying PSEs (none, rand, LapPE, RWSE, AllPSE) as well as GPSE, where as per usual we use a pre-trained GPSE node encoder trained on MolPCBA, on ZINC-12k. The exact architectures, like the number of layers and embedding sizes of the downstream models, vary based on previous studies. We refer to the human-readable YAML files in our code and the code of Cantürk et al. (2024), for the exact architectures. As in Cantürk et al. (2024), GCN, GatedGCN, and GIN are trained using Adam with a learning rate of $10^{-3}$, a weight decay of $10^{-5}$, for 1000 epochs with early stopping based on a validation split. Additionally, they use PyTorchs ReduceLROnPlateau to multiply the learning rate with 0.5 if the loss does not reduce for 10 consecutive epochs. GPS on other hand uses AdamW with the same learning rate and weight decay, but trained for 2000 epochs with the learning rate modified by

Table 9: GPSE processing encoder hyperparameters for MoleculeNet small benchmarks and synthetic WL graph benchmarks.

(a) MoleculeNet small benchmarks settings.

| Hyperparameter | |
| --- | --- |
| PE dim | 64 |
| PE encoder | Linear |
| Input dropout | 0.30 |
| Output dropout | 0.10 |
| Batchnorm | yes |
| Learning rate | 0.003 |
| # Epochs | 100 |
| # Warmup epochs | 5 |
| Weight decay | 0 |

(b) Synthetic WL graph benchmarks settings.

| Hyperparameter | |
| --- | --- |
| PE dim | 128 |
| PE encoder | Linear |
| Input dropout | 0.00 |
| Output dropout | 0.00 |
| Batchnorm | yes |
| Learning rate | 0.002 |
| # Epochs | 200 |
| Weight decay | 0 |

PyTorchs CosineAnnealingWarmRestarts with 50 warmup epochs a typical choice for transformer models. For Table 14 we collect for each backbone $B$ using each PSE (except GPSE) $P$ the number of epochs $e_B^P$ until the best validation performance is achieved as well as the test performance at this point. We then compute how many epochs GPSE takes to reach this same test performance $e_B^{\text{GPSE}>P}$. We report $\frac{e_B^P}{e_B^{\text{GPSE}>P}}$ that is the speedup of GPSE over $P$.

### E.2 Influence of Sample Size

This experiment is modeled after the experiment in Table 3 and Table 4 of Cantürk et al. (2024) considering only the GPS backbone. The GPS backbone is trained with varying PSEs (none, rand, LapPE, RWSE, AllPSE) as well as GPSE, GPSE$^-$, and GPSE$^+$ on ZINC-12k. Also, a GINE backbone is trained with varying PSEs (LapPE, RWSE, AllPSE) as well as GPSE, GPSE$^-$, and GPSE$^+$ on MolNet. All of these models are trained for a varying fraction of the dataset. The base case for this is a split of 80% for the training set, 10% for the validation set, and 10% for the test set. This 80% for the training set is then progressively reduced by factors of 2 without changing the size of the validation and test sets. The smallest training set fraction is $1.25\% = 80\%\frac{1}{2^6}$ for a total of 7 different training set fractions. We average each of these runs over 10 random seeds. Finally, we subtract from each method except GPSE the performance of GPSE for the respective training set fraction and report the results in Fig. 2 and Fig. 3 including their standard deviation.

### E.3 Generalization of PSEs

This experiment is almost the same as Table 2. The only difference is that we do not report the speedup but instead report the MAE on the test set in Table 16 as well as the train set in Table 5. In addition, we also report the difference of the two in Table 3. Following the experiment from Fig. 2 and Fig. 3 we also conduct this experiment for the MolNet datasets training as before a GINE backbone instead of a GPS backbone for a slightly different set of PSEs. We present the AUROC on the test set in Table 17 and the difference between the AUROC on the test and train set in Table 4.

## F Datasets

A collection of task information and classical graph properties for all considered datasets are presented in Table 10 and Table 11. What follows are short descriptions and citations for each of the used datasets.

**MolPCBA** (Hu et al., 2020) (MIT License) contains 400K small molecules derived from the MoleculeNet benchmark (Wu et al., 2017). There are 323,555 unique molecular graphs in this dataset.

Table 10: Task information for datasets used in transferability experiments. TRIX* shows the information for the test set of TRIX.

| Dataset | Num. graphs | Num. nodes | Num. edges | Pred. level | Pred. task | Num. tasks | Metric |
|---------|------------|-----------|-----------|------------|-----------|-----------|--------|
| ZINC-12k | 12,000 | 23.15 | 24.92 | graph | reg. | 1 | MAE |
| MolHIV | 41,127 | 25.51 | 27.46 | graph | class. (binary) | 1 | AUROC |
| MolPCBA | 437,929 | 25.97 | 28.11 | graph | class. (binary) | 128 | AP |
| MolBBBP | 2,039 | 24.06 | 25.95 | graph | class. (binary) | 1 | AUROC |
| MolBACE | 1,513 | 34.09 | 36.86 | graph | class. (binary) | 1 | AUROC |
| MolTox21 | 7,831 | 18.57 | 19.29 | graph | class. (binary) | 21 | AUROC |
| MolToxCast | 8,576 | 18.78 | 19.26 | graph | class. (binary) | 617 | AUROC |
| MolSIDER | 2,039 | 33.64 | 35.36 | graph | class. (binary) | 27 | AUROC |
| CSL | 150 | 41.00 | 82.00 | graph | class. (10-way) | 1 | ACC |
| EXP | 1,200 | 48.70 | 60.44 | graph | class. (binary) | 1 | ACC |
| CEXP | 1,200 | 55.78 | 69.78 | graph | class. (binary) | 1 | ACC |
| TRI | 1,000 | 20.00 | 30.00 | node | class. (binary) | 1 | ACC |
| TRIX* | 1,000 | 100.00 | 150.00 | node | class. (binary) | 1 | ACC |

Table 11: Classical graph properties of graph-level datasets used in transferability experiments. TRIX* shows the classical graph properties for the test set of TRIX.

| | Num. nodes | Num. edges | Density | Connectivity | Diameter | Approx. max clique | Centrality | Cluster. coeff. | Num. triangles |
|---|-----------|-----------|---------|-------------|----------|-------------------|-----------|----------------|---------------|
| ZINC-12k | 23.15 | 24.92 | 0.101 | 1.00 | 12.47 | 2.06 | 0.101 | 0.006 | 0.06 |
| MolHIV | 25.51 | 27.46 | 0.103 | 0.927 | 11.06 | 2.02 | 0.103 | 0.002 | 0.03 |
| MolPCBA | 25.97 | 28.11 | 0.093 | 0.998 | 13.56 | 2.02 | 0.093 | 0.002 | 0.02 |
| MolBBBP | 24.06 | 25.95 | 0.114 | 0.950 | 10.75 | 2.03 | 0.114 | 0.003 | 0.03 |
| MolBACE | 34.09 | 36.86 | 0.070 | 1.00 | 15.22 | 2.10 | 0.070 | 0.007 | 0.10 |
| MolTox21 | 18.57 | 19.29 | 0.157 | 0.976 | 9.37 | 2.02 | 0.159 | 0.003 | 0.03 |
| MolToxCast | 18.78 | 19.26 | 0.154 | 0.803 | 7.57 | 2.02 | 0.154 | 0.003 | 0.03 |
| MolSIDER | 33.64 | 35.36 | 0.103 | 0.856 | 12.45 | 2.02 | 0.120 | 0.004 | 0.04 |
| CSL | 41.00 | 82.00 | 0.100 | 3.98 | 6.00 | 2.10 | 0.100 | 0.050 | 4.10 |
| EXP | 48.70 | 60.44 | 0.054 | 0.00 | 1.00 | 2.00 | 0.054 | 0.000 | 0.00 |
| CEXP | 55.78 | 69.78 | 0.047 | 0.19 | 4.03 | 2.00 | 0.047 | 0.000 | 0.00 |
| TRI | 20.00 | 30.00 | 0.158 | 2.62 | 5.14 | 2.84 | 0.158 | 0.088 | 1.75 |
| TRIX* | 100.00 | 150.00 | 0.030 | 2.94 | 8.76 | 2.81 | 0.030 | 0.017 | 1.66 |

To extract unique molecular graphs, we use RDKit with the following steps:

1. For each molecule, convert all its heavy atoms to carbon and all its bonds to single-bond.

2. Convert the modified molecules into a list of SMILES strings.

3. Reduce the list to unique SMILES strings using the `set()` operation in Python.

**ZINC-12k** (Dwivedi et al., 2022c) (Custom license, free to use) is a 12K subset of the ZINC250K dataset (Gómez-Bombarelli et al., 2018). Each graph is a molecule whose nodes are atoms (28 possible types) and whose edges are chemical bonds (3 possible types). The goal is to regress the constrained solubility (Dwivedi et al., 2022c) (logP) of the molecules. This dataset comes with a pre-defined split with 10K training, 1K validation, and 1K testing samples.

**MoleculeNet small datasets** (Hu et al., 2020) (MIT License) We follow Sun et al. (2022) and use the selection of five small molecular property prediction datasets from the MoleculeNet benchmarks, including BBBP, BACE, Tox21, ToxCast, and SIDER. Each graph is a molecule, and it is processed the same way as for MolHIV and MolPCBA. All these datasets adopt the *scaffold splitting* strategy that is similarly used on MolHIV and MolPCBA.

**CSL** (Dwivedi et al., 2022c) (MIT License) contains 150 graphs that are known as circular skip-link graphs (Murphy et al., 2019). The goal is to classify each graph into one of ten isomorphism classes. The dataset is class-balanced, where each isomorphism class contains 15 graph instances. Splitting is done by stratified five-fold cross-validation.

**EXP and CEXP** (Abboud et al., 2021) (*unknown* license) contain 600 pairs of graphs (1,200 graphs in total) that cannot be distinguished by 1&2-WL tests. The goal is to classify each graph into one of two isomorphism classes. Splitting is done by stratified five-fold cross-validation. CEXP is a modified version of EXP, where 50% of pairs are slightly modified to be distinguishable by 1-WL.

**TRI and TRIX** (Sato et al., 2021) (*unknown* license) contain 1000 3-regular 20-order graphs. The goal is to classify for each node whether it is contained in a triangle. Since the node colors of regular graphs are stable under 1-WL, MPNNs cannot tell nodes in these graphs apart, and thus, this task cannot be solved by vanilla MPNNs. TRIX is an extrapolation dataset of TRI, that shares the exact same training (and validation) dataset, however, the testset contains 1000 3-regular 100-order graphs. Splitting is done by five-fold cross-validation.

# G GPSE Convergence to PSEs Performance

Here we present additional results showing the speedup GPSE$^-$(Table 12) and GPSE$^+$(Table 13) offer compared to the baseline PSEs instead of GPSE on ZINC-12k.

We also present the average epoch that early stopping stopped on for the GPSE variants and all PSEs in Table 14. This table shows that GPSE and its variants do not always converge faster to their respective optima when compared to other PSEs.

Further, we also present the speedup GPSE offers compared to baseline PSEs on MolNet in Table 15. Note that since GPSE does not always increase the overall performance, there are cases in which GPSE never reaches a better value than the baseline PSE. Thus, we mark each entry with an exponent indicating the number of seeds on which GPSE outperformed the baseline PSE and the statistics for these seeds. If GPSE performs better on at least one seed, the average and standard deviation are shown. If there is no such seed, the table reads inf.

# H Influence of Sample Size on MolNet

Fig. 3 shows the influence of the size of the training dataset on performance for all MolNet datasets. In the main body of this work, we presented the figure for ToxCast, which is arguably the worst case for GPSE, as

Table 12: Using GPSE$^-$on ZINC-12k significantly reduces the number of epochs needed to reach the same test performance as other PSEs. This table shows the factor of speedup (in terms of epochs) of GPSE$^-$to reach the same MAE on ZINC-12k on various downstream GNNs and PSEs. For instance, 2 means GPSE$^-$reaches the same performance with only half of the epochs.

| PSE↓ \ Downstream→ | GCN | GatedGCN | GIN | GPS |
|---|---|---|---|---|
| none | 68.3±8.3 | 68.8±10.1 | 58.2±12.0 | 11.9±1.5 |
| rand | 905.0±0.0 | 110.0±0.0 | 253.0±0.0 | 95.0±16.1 |
| LapPE | 46.5±13.0 | 47.4±7.6 | 33.2±3.3 | 14.1±1.8 |
| RWSE | 20.6±2.2 | 26.4±2.5 | 21.4±4.5 | 2.7±0.3 |
| AllPSE | 13.4±1.2 | 18.7±1.7 | 18.9±2.5 | 2.6±0.2 |

Table 13: Using GPSE$^+$on ZINC-12k significantly reduces the number of epochs needed to reach the same test performance as other PSEs. This table shows the factor of speedup (in terms of epochs) of GPSE$^+$to reach the same MAE on ZINC-12k on various downstream GNNs and PSEs. For instance, 2 means GPSE$^+$reaches the same performance with only half of the epochs.

| PSE↓ \ Downstream→ | GCN | GatedGCN | GIN | GPS |
|---|---|---|---|---|
| none | 68.9±13.5 | 66.1±12.7 | 56.9±10.2 | 11.6±1.3 |
| rand | 905.0±0.0 | 110.0±0.0 | 227.7±50.6 | 84.4±17.2 |
| LapPE | 46.1±7.5 | 45.8±3.7 | 33.7±4.7 | 13.7±1.6 |
| RWSE | 20.6±3.7 | 29.8±3.1 | 20.4±3.0 | 2.8±0.4 |
| AllPSE | 13.0±2.2 | 21.1±3.3 | 18.7±2.5 | 2.6±0.2 |

Table 14: GPSE variants do not actually converge faster than other methods in optimal performance. This table shows the epoch on which early stopping stopped according to MAE on ZINC-12k on various downstream GNNs and PSEs.

| PSE↓ \ Downstream→ | GCN | GatedGCN | GIN | GPS |
|---|---|---|---|---|
| none | 827.7±201.4 | 591.2±348.4 | 863.8±221.9 | 1304.9±311.5 |
| rand | **667.1±219.9** | **189.7±114.9** | **386.5±206.0** | **167.0±34.4** |
| LapPE | 669.9±239.2 | 628.6±256.8 | 838.3±108.1 | 1070.3±466.2 |
| RWSE | 934.5±75.1 | 410.3±281.1 | 630.7±255.5 | 1373.2±194.9 |
| AllPSE | 715.7±169.6 | 425.7±252.0 | 506.4±316.9 | 1543.8±256.4 |
| GPSE | 935.7±66.9 | 803.9±163.1 | 919.2±55.2 | 1238.2±224.9 |
| GPSE$^-$ | 939.1±68.4 | 875.5±108.4 | 933.9±72.4 | 1497.5±243.8 |
| GPSE$^+$ | 934.1±44.0 | 857.0±171.7 | 910.8±119.4 | 1426.4±166.6 |

Table 15: No clear improvements can be seen for GPSE on the MolNet datasets with regard to the number of epochs needed to reach the same test performance as other PSEs. This table shows the factor of speedup (in terms of epochs) of GPSE to reach the same AUC on various MolNet datasets and PSEs. For instance, 2 means GPSE reaches the same performance with only half of the epochs. The exponent indicates the number of seeds for which GPSE achieved a better performance than the respective PSE. inf indicates that there is no seed for which GPSE outperforms the baseline PSE.

| PSE ↓ \ Dataset→ | BACE | BBBP | ClinTox | HIV | MUV | SIDER | Tox21 | ToxCast |
|---|---|---|---|---|---|---|---|---|
| GraphLog | inf | inf | inf | inf | inf | $4.2\pm0.5^{10}$ | $1.2\pm0.1^{3}$ | $2.1\pm0.5^{10}$ |
| LapPE | $17.8\pm3.3^{10}$ | inf | inf | inf | inf | $5.8\pm0.9^{10}$ | inf | inf |
| RWSE | $20.6\pm2.2^{10}$ | inf | inf | inf | inf | $6.3\pm0.7^{10}$ | inf | inf |
| AllPSE | $3.9\pm0.7^{10}$ | inf | inf | inf | $0.9\pm0.1^{10}$ | $2.6\pm0.3^{8}$ | inf | inf |

Table 16: GPSE and its variants always perform better than other PSEs regardless of the backbone on ZINC-12k. In only one case, that is, using the GPS backbone, AllPSE performs only insignificantly worse than the GPSE variants. This table shows the test MAE (↓) on ZINC-12k for various backbone networks and PSEs.

| PSE ↓ \ Downstream→ | GCN | GatedGCN | GIN | GPS |
|---|---|---|---|---|
| none | 0.289±0.008 | 0.244±0.011 | 0.278±0.013 | 0.123±0.01 |
| rand | 1.255±0.027 | 1.221±0.008 | 1.242±0.015 | 0.87±0.014 |
| LapPE | 0.218±0.008 | 0.189±0.007 | 0.217±0.005 | 0.141±0.078 |
| RWSE | 0.177±0.004 | 0.165±0.003 | 0.175±0.005 | 0.075±0.005 |
| AllPSE | 0.152±0.003 | 0.142±0.005 | 0.15±0.004 | 0.071±0.009 |
| GPSE | **0.125±0.003** | **0.112±0.003** | **0.127±0.003** | 0.07±0.005 |
| GPSE⁻ | 0.128±0.003 | 0.114±0.003 | 0.129±0.003 | 0.067±0.004 |
| GPSE⁺ | 0.127±0.002 | 0.113±0.002 | 0.128±0.002 | **0.066±0.003** |

Table 17: GPSE or one of its variants frequently perform best or only insignificantly worse than other PSEs. This table shows the test AUROC(↑) on MolNet datasets for different PSEs, with a GINE downstream network.

| PSE ↓ \ Dataset→ | BACE | BBBP | ClinTox | HIV | MUV | SIDER | Tox21 | ToxCast |
|---|---|---|---|---|---|---|---|---|
| GraphLog | 82.6±1.6 | 66.8±2.1 | 73.4±3.9 | 75.0±2.0 | 74.9±1.3 | 60.6±0.8 | 73.4±0.9 | 63.0±0.8 |
| LapPE | 78.1±2.6 | **67.7±2.8** | 75.4±2.8 | 76.2±1.5 | 75.0±1.2 | 60.7±0.9 | 77.4±0.4 | 64.5±0.7 |
| RWSE | 81.4±2.2 | 66.3±2.0 | 73.4±4.0 | **78.6±1.0** | **77.1±0.9** | 59.2±0.8 | 76.0±0.7 | 63.9±0.5 |
| AllPSE | 77.9±2.8 | 66.8±0.7 | 72.0±3.5 | 74.7±1.7 | 66.9±0.5 | **62.9±0.3** | 76.0±0.6 | 64.1±0.4 |
| GPSE | **83.1±1.6** | 67.3±1.0 | **76.7±4.6** | 76.1±2.1 | 75.3±0.8 | 62.7±1.2 | 77.2±0.6 | 65.9±0.7 |
| GPSE⁻ | 81.6±1.2 | 67.6±1.1 | 76.5±4.9 | 76.7±1.9 | 76.2±1.7 | 62.2±1.4 | **77.5±0.9** | 65.7±0.6 |
| GPSE⁺ | 80.0±2.3 | 67.3±1.5 | 76.7±2.4 | 77.5±1.4 | 76.0±1.3 | 61.6±2.6 | 77.2±0.6 | **66.0±0.6** |

for a smaller fraction of available training data, all PSEs overtake GPSE in performance. Tox21 is notably another good example of GPSE, as the difference in performance grows for a smaller training data ratio. In combination, these figures indicate that GPSE's performance in a data-scarce regime depends on the dataset in question, even though GPSE is better than other PSEs in most cases.

# I  Generalization of PSEs

Here we show the test performance in relation to Table 3 and Table 4. As already shown in Cantürk et al. (2024), Table 16 and Table 17 show the test performance of various PSEs against various downstream backbones (on ZINC-12k showing MAE) and various MolNet datasets (showing AUROC). Notably, GPSE and its variants perform very well and usually either have the best performance or are only insignificantly worse than the best performance. Notably, for only 3 datasets (BACE, HIV, and MUV), some GPSE variants don't belong to the best performers.

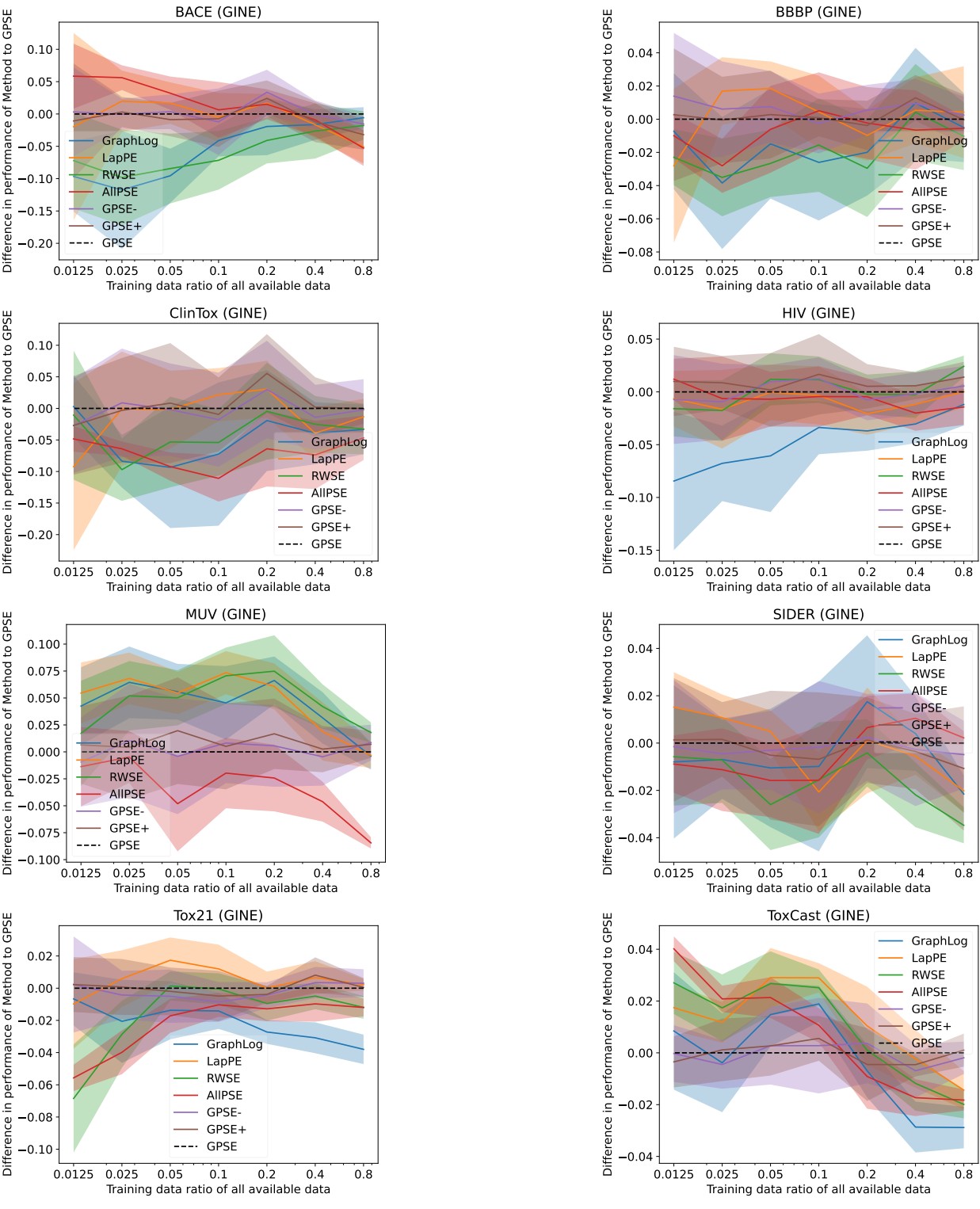

Figure 3: In most cases, GPSE variants outperform other PSEs regardless of available training data on the MolNet datasets. There are, however, also examples where GPSE is outperformed by other PSEs for less available training data. Results on additional datasets beyond what is shown in Fig. 2. Downstream training with fractions of the MolNet datasets. We show the difference in performance (AUROC ↑) obtained by various PSEs, including GPSE⁻and GPSE⁺.

