# OpenReview forum: "Towards Graph Foundation Models: A Study on the Generalization of Positional and Structural Encodings"
_TMLR — Accepted by TMLR_

### Review · Reviewer_zGyi · 2024-12-30

**Summary Of Contributions:**

Recent advances in integrating positional and structural encodings (PSEs) into graph neural networks (GNNs) have significantly enhanced their performance across various graph learning tasks. However, the general applicability of these encodings and their potential to serve as foundational representations for graphs remain uncertain. This paper investigates the fine-tuning efficiency, scalability with sample size, and generalization capability of learnable PSEs across diverse graph datasets.

**Audience:**

Yes

**Broader Impact Concerns:**

No concerns.

**Claims And Evidence:**

Yes

**Requested Changes:**

Please see the weaknesses above. I'm not an expert in this field, so detailed explanation/intuition would be helpful.

**Strengths And Weaknesses:**

Strengths
1. This work studies theoretical expressivity of GPSE.
2. This work performs various experiments on GPSE, focusing on its convergence speed or generalization ability.
3. This work introduces variants of GPSE, which improve the practical applicability of the method.

Weaknesses/questions
1. Since this work studies GPSE as the main target, there should be more insights on why GPSE is worth studying. To my understanding, GPSE is just a GNN trained to mimic PSEs, and I’m not convinced why we should use GPSE instead of the original PSEs despite its higher computational cost.
2. Theorem 1 is an important contribution, but being able to approximate GIN is different from being able to compute “any” PSEs. There should be comprehensive analysis on the expressivity of PSEs to make such a claim, e.g., whether all PSEs can be approximated by GIN or not.
3. The term “downstream universality” is not well defined. Test performance is often not alined with theoretical expressivity, and that is natural. The authors show in Table 1 actual performances of models on various datasets, but they may not be an appropriate way to report the expressivity of encodings for downstream tasks.
4. Related to W1, I’m not sure why GPSE is able to achieve faster convergence (in Table 2) if they are trained to mimic PSEs. Is it because they work as a smooth approximator that helps models get better signals of the graph structure? More intuitions would be helpful.

---

> ### Author Response · Authors · 2025-01-16
> **Rebuttal to Reviewer zGyi (1)**
>
> We sincerely thank the reviewer for their thoughtful feedback and for recognizing the strengths of our work. Below, we address the concerns raised in the review. We have uploaded an updated manuscript with all changes marked in red.
> ## Regarding Weakness 1: Motivation for Studying GPSE
> We agree that it is important to clarify why GPSE is worth studying. GPSE is interesting primarily because training the GPSE model to mimic/reconstruct the original PSEs gives rise to a model that outperforms not only any individual PSE, but also combinations and concatenations of multiple PSEs, across a large variety of benchmarks. This in itself demonstrates that GPSE does more than simply mimic the original PSEs, contrary to the reviewer’s argument. This is discussed in more detail both in the original paper (Appendix I) and in this rebuttal (please see our response to Weakness 4),but GPSE goes beyond mimicking the original PSEs by distilling the signals coming from them to arrive at a robust and unified encoding.
>
> Additionally, as mentioned in both our work and the original GPSE paper, the GPSE model is used in pre-trained form to generate PSE representations, this makes it very cheap and fast since all we perform is a forward pass through our GNN. PSEs have no inherent transferability, and thus need to be computed for every graph (which for many PSEs has a significantly high cost and does not scale well); in Figure G.1 of the original GPSE paper, we see that this GNN forward pass is cheaper to compute than any standalone PSE computation while outputting an encoding that encapsulates all PSEs the original model has been trained on. The GPSE paper points out that computing all PSEs is orders-of-magnitude slower than the GPSE forward pass, so the reviewer’s claim of “higher computational cost” is invalid (at least for a pre-trained GPSE in inference mode, which is how GPSE generates encodings that function as PSE replacements). We again note that GPSE encodings consistently outperform the concatenation of all PSEs (AllPSE), and thus GPSE encodings are both more performative and cheaper to generate.
>
> In short, our strong conviction is that GPSE represents a novel and interesting direction in graph learning: As demonstrated by both the original paper and our work, it (a) provides clear practical improvements both in terms of performance and efficiency compared to conventional PSEs, and (b) poses interesting theoretical questions in how training a GNN to “mimic” PSEs can give rise to encodings that improve on all PSEs trained on, and promotes further discussion on how such phenomena can help us develop graph foundational models, which we aimed to tackle in this work.
> ## Regarding Weakness 2: Expressivity of GIN and PSEs
> We acknowledge that the statement of this weakness is correct—GIN cannot compute all PSEs. For example, consider the PSE that computes the orbit partition of a graph (assigning nodes that are not isomorphic different colors). This cannot be computed by GIN for many 3-regular graphs. Specifically, GIN must assign every node the same color for 3-regular graphs, yet there exist 3-regular graphs with nodes that are not equivalent.
>
> For instance, consider the tetrahedron graph (the unique 3-regular graph on 4 nodes) and any 3-regular graph on 6 nodes. These two graphs are not isomorphic, so their nodes are not equivalent. Thus, the disjoint union of these two graphs provides an example where GIN cannot compute the orbit partition.
>
> That said, it is important to note a key conclusion from prior research: **any non-random PSE can be approximated by GIN with random node features (RNF)**. This is a critical observation, as it highlights the distinction between approximating deterministic PSEs and random functions, the latter of which no model can reasonably be expected to approximate. We have notably already included such a statement on page 4 section 4.1 “Furthermore, Theorem 4 from Franks et al. (2023) [...] proves that, indeed, under mild assumptions, GPSE is universal and can learn to embed any PSE.” Regarding the last sentence, we are unsure how a detailed analysis of the expressivity of PSEs could fit the scope of this work, since such an analysis would be a challenging undertaking for most PSEs.

---

> ### Author Response · Authors · 2025-01-16
> **Rebuttal to Reviewer zGyi (2)**
>
> ## Regarding Weakness 3: Downstream Universality
> We added clarifications for the term "downstream universality" in the revised manuscript to the notations and background section. By this term, we mean the following: **whether a pretrained GPSE model, used as a fixed PSE after pretraining, provides universality to the downstream model**. More specifically, this addresses the question:
> *"Can GIN using GPSE as a PSE (where the GPSE model is frozen) approximate any function?"*
>
> This is a challenging question, as the answer depends on the exact parameters of GPSE. Table 1 provides empirical justification for the belief that GPSE, as trained in prior work, does not grant universality. Additionally, Theorem 2 provides a theoretical statement that under reasonable assumptions—namely, that the output of GPSE does not contain randomness—GPSE cannot provide universality. This answers the question in the negative under this assumption.
> ## Regarding Weakness 4: Faster Convergence of GPSE
> At a high level, we can think of GPSE training as learning a well conditioned and smooth low-dimensional data manifold, similar to the idea of a denoising autoencoder. The original GPSE paper provides some additional intuition on this in Appendix I:
>
> *Learning to encode a diverse collection of PSEs leads to a general embedding space that abstracts both local and global perspectives of the query graph, which are more readily usable by the downstream model compared to the unprocessed PSEs. This also explains why a joint encoding outperforms the concatenation of all encodings, AllPSE. Concatenating many encodings very likely to leads to redundant representations, and also introduces significant noise to node features particularly in datasets/downstream tasks where only a small portion of the encodings are useful: This is reflected in most experiments where not only GPSE, but also RWSE- or LapPE-only configurations outperform AllPSE.*
>
> Essentially, the reconstruction loss used in GPSE training is simply a proxy to learn useful joint latent representations that indeed helps the downstream model get better signals of the graph structure. With this in mind, it is a reasonable inference that learning on a smoother, denoised signal will indeed lead to faster convergence; in that sense, the reviewer’s intuition is correct.
>
> Lastly, we thank the reviewer for their valuable comments and suggestions. These insights will help us significantly improve the clarity and quality of our manuscript.

---

### Review · Reviewer_tBaA · 2025-01-07

**Summary Of Contributions:**

The authors examine GPSE and its generalization abilities in the context of graph foundation models. They also discuss the expressivity of GPSE and how it may affect the expressivity of downstream models. This work showcases comprehensive experiments examining GPSE's expressivity, convergence time, and generalization error. They also provide a more robust version of GPSE, GPSE+, which adds relevant "hard" graphs to GPSE's pre-training stage.

**Audience:**

Yes

**Claims And Evidence:**

Yes

**Requested Changes:**

Besides, addressing the above weaknesses, here are some requested changes.

- There is some missing discussion of [1]'s study on using biconnectivity to evaluate GNN expressivity.
- Can the authors include some discussion of the significance of GPSE, GPSE^+, and GPSE^-'s performance and it means for GFM practitioners in the wild? It's difficult to find a concrete take-home message for PSEs for GFMs from these empirical studies. Also, it seems counterintuitive for GPSE^- to perform better than its more expressive variants on TRIX.

[1] Zhang et al. Rethinking the Expressive Power of GNNs via Graph Biconnectivity. ICLR 2023.

**Strengths And Weaknesses:**

**Strengths**

- The paper is generally well-written.
- The role of PSEs for graph foundation models is underexplored, and it is good to see that people are working on this.
- The authors take a deeper dive compared to the original paper regarding GPSE's empirical performance and expressivity.

**Weaknesses**
- The theoretical contributions are rather modest. For instance, Theorem 1 and the discussion after it largely builds off of [1] and [2], and the points made essentially boil down to:

1. GPSE is built on GatedGCN -> GIN can simulate 1-WL [1], and GatedGCN can simulate GIN -> GPSE can simulate 1-WL

2. GPSE uses RNFs -> with sufficiently many layers of sufficient width, an MPNN with RNFs can compute any graph structure function [2] -> sufficiently wide and deep GPSE can compute any graph structure function

From this perspective, if I'm understanding correctly, 1. seems redundant if you have 2.

- It's not entirely clear to me why Theorem 2 implies that a random variable is required as output. More general and intuitive discussion as to why graphs A and B are 1-WL indistinguishable and why that's significant for PEs in general would be helpful.

- Organizationally, this work reads a bit disjointed, like two smaller works combined. One part studies the expressivity of GPSE while the other studies the generalizability of GPSE with little discussion bridging the two topics. As the paper admits, they do not seek to conflate expressivity and generalization, but elucidating the relationship between the two for GPSE seems to be the most natural step in tying together all of the paper's main contributions. This does not hurt correctness, and the paper seems relatively modest about what it claims, but I think it does detract from its overall quality.

[1] Xu et al. How Powerful are Graph Neural Networks? ICLR 2019.

[2] Franks et al. A systematic approach to universal random features in graph neural networks. TMLR, 2023.

---

> ### Author Response · Authors · 2025-01-16
> **Rebuttal to Reviewer tBaA**
>
> We sincerely thank the reviewer for their detailed and thoughtful feedback, as well as for acknowledging that our work is mostly correct and addressing an important and underexplored area of research. Your comments have provided us with valuable insights, and we deeply appreciate the recognition of our efforts to investigate GPSEs and their role in graph foundation models. We have uploaded an updated manuscript with all changes marked in red.
> ## Clarification on Theoretical Contributions (First Bullet Point in Weaknesses)
> We acknowledge that your summary of the statements in 1 and 2 is correct. However, we would like to note that we believe that Theorem 1 is not redundant given Theorem 2. To be specific, for RNF-GNNs to be universal, they must be able to simulate the 1-WL test. Without Theorem 1, it would remain uncertain whether GatedGCNs can simulate 1-WL. Indeed, Theorem 1 explicitly ensures that this simulation capability exists, which is critical to this overall result.
> ## Clarification on Theorem 2 (Second Bullet Point in Weaknesses)
> Thank you for the thorough comment. Theorem 2 assumes the strongest possible non-random positional encoding, specifically, that the orbit partition is provided as input. To be more precise, in order to compute a coloring that is coarser than the orbit partition, an ordering of the nodes is required. This ordering typically necessitates some form of randomness. For instance, if two nodes belong to the same orbit (i.e., there exists an isomorphism mapping one node to the other, rendering them equivalent), they must be assigned the same color unless there exists an explicit node ordering that can be used to distinguish them. Thus, Theorem 2 states that using the “strongest possible” PSE (in a coloring sense) is not enough for universality, i.e., even GPSE assuming it computes at most the orbit partition (no ordering/randomness remaining) is not universal.
>
> Regarding your specific question, we would like to clarify whether you are referring to graphs (a) and (b) from Figure 1, or to graphs A (c in Figure 1) and B (d in Figure 1). It should be evident that both pairs of graphs are distinct. However, they cannot be distinguished by the 1-WL test because they are already stable under 1-WL and:
> - In (a) and (b), all black nodes have exactly two black neighbors.
> - In (c) and (d), all black nodes have two red and two black neighbors, and all red nodes have two black and two red neighbors. Notably, red/black nodes can be distinguished from one another due to their initial coloring.
>
> These statements are sufficient to conclude that the graph pairs (a), (b) and (c), (d) cannot be distinguished since after the recoloring step, we only check whether both graphs in a pair have the same color partition, i.e., the same amount of nodes of each color. This partition is six black nodes for (a) and (b). This partition is six black and red nodes for (c) and (d).
>
> We hope this clarification helps address your concern. We would like to kindly ask the Reviewer to clarify their question, in case we misunderstood it, and we will be happy to provide further details.
> ## Organizational Comments (Last Bullet Point in Weaknesses)
> Thank you for the comment. We note here, if the paper is viewed as a more detailed review of GPSE, then it naturally follows that we would take a closer look at its expressivity. In this context, we felt it was essential to study the expressivity of GPSEs and positional encodings (PSEs) in general alongside the generalization results. We acknowledge your feedback regarding the perceived disjointedness and will consider better ways to connect these topics in future work.
> ## Regarding Requested Changes
> - Discussion of Biconnectivity and Expressivity:
> We appreciate this suggestion and have added further discussion regarding finer expressivity metrics in Appendix A. Specifically, we have addressed how methods can be further categorized or ordered based on their performance on metrics such as biconnectivity and homomorphism counting.
> - Practical Implications for GPSE, GPSE+, and GPSE-:
> While the current version of the paper already includes some discussion on this point, we acknowledge that more detailed insights could be helpful. We  have included additional commentary on the significance of GPSE, GPSE+, and GPSE-'s performance for practitioners.
>
> Thank you once again for your valuable feedback, which has helped us identify areas for improvement. We are committed to addressing your comments and suggestions to enhance our work's clarity, rigor, and practical relevance.

---

> > ### Comment · Reviewer_tBaA · 2025-01-22
> >
> > Thank you for the response and revised paper.
> >
> > **Theoretical contributions**
> >
> > I think it would be helpful to explain what's meant by a "function on the graph structure" in the context of Theorem 1. I'm not familiar with Theorem 4 from Franks et al. (2023), so it would be helpful to explain why 1-WL is distinct from this. To my knowledge, 1-WL is a graph isomorphism test, so it should fall under the umbrella of graph structure functions if "graph structure function" is a general term for functions that operate on graphs.
> >
> > **Theorem 2**
> >
> > I was referring to graphs (c) and (d) in Figure 1. Thank you for the clarification. I think that first paragraph on orbit partitions is what I was looking for and makes it more clear why random features are necessary and how it relates to orbit partitions.
> >
> > **Requested Changes**
> >
> > Thank you for the changes. I think making the practical implications as explicit as possible is helpful in forming a concrete take-home message for the paper.

---

> > > ### Author Response · Authors · 2025-01-23
> > > **Explaining "function on the graph structure"**
> > >
> > > Thank you once again for your comments.
> > >
> > > We agree that it could be beneficial to specify the statement a "function on the graph structure" and have thus revised our manuscript, adding "(that is functions of the form $f((V,E))$ ignoring node/edge features)" to this statement. This change is also marked in red.
> > >
> > > Please feel free to voice any further concerns so we can address them.

---

### Review · Reviewer_qSbh · 2025-01-13

**Summary Of Contributions:**

The article explores the role of positional and structural encodings (PSEs) in Graph Neural Networks (GNNs) and their potential as foundational components for graph-based tasks. Building on the recently proposed GPSE model, the authors introduce and analyze two variants, GPSE+ and GPSE−. The study focuses on three key aspects: expressivity, scalability with varying sample sizes, and generalization performance across diverse datasets.

While the article is of fundamental interest to the community I found the read a bit complex, overall I would suggest clarifying some aspects defined bellow.

**Audience:**

Yes

**Broader Impact Concerns:**

No concerns.

**Claims And Evidence:**

Yes

**Requested Changes:**

1. It is unclear why only 4-regular graphs are used in GPSE+. While the choice of the number of nodes aligns with the average graph size in the MolPCBA dataset, there are many WL-indistinguishable sets of graphs. Why were 4-regular graphs specifically chosen?
2. The article relies heavily on the GPSE model. I suggest providing a more detailed explanation of it in the main text, rather than relegating it to the supplementary material.
3. Several key concepts (e.g., LapPE, AllPSE) are explained only in the supplementary material. It would be helpful to include brief explanations of these in the main text for better clarity.
4. Could you clarify the "mild assumptions" referenced in Theorem 1? Providing explicit details would make the theorem easier to understand.
5. Tables and figures are positioned far from the text that discusses them (e.g., Table 1, Figure 1), making the paper harder to follow. Consider reorganizing to improve readability.

**Strengths And Weaknesses:**

**Strengths**

1. The article addresses a critical challenge in machine learning on graphs, effectively linking advancements in GNNs with the emerging concept of Graph Foundation Models (GFMs).
2. It provides a thorough evaluation of GPSE and its variants (GPSE+, GPSE−) across both synthetic and real-world datasets, offering valuable insights into scalability and generalization.

**Weaknesses**
1. The article is somewhat difficult to follow for non-expert readers, as it requires frequent back-and-forth reference between the main text and the supplementary materials.
2. Some decisions are not well motivated in the text.

---

> ### Author Response · Authors · 2025-01-16
> **Rebuttal to Reviewer qSbh**
>
> We thank the reviewer for their thoughtful comments, valuable suggestions, and for recognizing the strengths of our work. We are pleased that the article was found to address critical challenges in graph machine learning. Below, we respond to the requested changes. We have uploaded an updated manuscript with all changes marked in red.
> ## Requested Change 1: Why Only 4-Regular Graphs in GPSE+?
> Thank you for the question. In our submission, we stated on page 6, Section 4.2, under GPSE+: "Specifically, relevant hard graphs for the CSL dataset are 4-regular graphs.". We mentioned this, because in the original GPSE paper, the performance on CSL graphs were poor, despite the expectations that pretraining GPSE against the cycle-counting PSE should make CSL perfectly solvable. We conjecture that the poor performance occurred because GPSE did not effectively learn to utilize the random node features provided for challenging graphs. Moreover, we note that CSL graphs are cycles with skip links, a subset of 4-regular graphs.
> Thus, we aim to demonstrate that including relevant hard graphs can enhance GPSE's performance. However, we are not asserting that using 4-regular graphs is the only possible or interesting choice, and we agree with the reviewer. To accommodate that point, we have revised the manuscript accordingly.
> ## Requested Change 2: Detailed Explanation of GPSE in the Main Text
> We agree with the suggestion and have moved the GPSE description to the paper's main body. This adjustment should improve clarity and make the text more self-contained. Thank you.
> ## Requested Change 3: Explanation of LapPE and AllPSE
> We agree that a brief description of the PSEs used in our work would be helpful in the main body. To this end, we have included concise explanations of LapPE, AllPSE, and other relevant PSEs to enhance clarity and accessibility for readers.
> ## Requested Change 4: Clarification of "Mild Assumptions" in Theorem 1
> Thank you for the question. The "mild assumptions" is vague in our submission because multiple reasonable assumptions can be made, as discussed in the last paragraph of the proof for Theorem 1 in Appendix C. Therefore, we prefer to leave the statement as it is to avoid enforcing a single specific assumption. However, if this request remains after this rebuttal, we will modify the text and choose to explicitly state the assumption that each node carries at least one constant 1 feature and that the norm of node features is bounded.
> ## Requested Change 5: Positioning of Tables and Figures
> We moved Table 1 and Figure 1 to more appropriate locations within the paper to improve readability and ensure they appear closer to the corresponding text. We think that it enhanced the readability of our paper – thank you.
>
> Lastly, we thank the reviewer for their valuable feedback and suggestions, which we believe helped improve the quality and clarity of the paper.

---

### Author Response · Authors · 2025-01-16
**Thank You**

We thank the reviewers for their time and efforts. We respond to each reviewer individually below. Notably, we have made changes to the manuscript in response to reviewer feedback. Most changes are marked in red text. We would be happy to respond to any additional questions.

---

### Decision · Action_Editor_9AiT · 2025-02-18

**Recommendation:** Accept as is

**Comment:**

The paper provides a thorough evaluation of GPSE and variants across multiple datasets, providing insights in terms of scalability and efficiency. This is a more in depth exploration of GPSE (compared to the original paper introducing it) and is accompanied by theoretical results on the expressivity of these embeddings. During the rebuttal period the authors answered the questions raised by the reviewers, and improved the write-up. All reviewers agreed that the quality of the paper improved,
Given the thoroughness of the paper reviewers recommend accept, which I think reflects the scholarly character of the work.

**Audience:**

The paper would be of interest to the GNN community, providing additional insights in positional and structural encodings (PSE). I believe anyone in the community exploring variants of PSEs will find the work very insightful, and the scholarly and careful evaluation of the different variants of PSE can be very helpful in guiding people in their research.

**Claims And Evidence:**

The paper looks at the fine-tuning efficiency and generalization of learnable PSEs across diverse graph structures. The reviewers agree that the paper is technically correct and thorough. During the rebuttal period the paper had seen updates that improve clarity and the flow of the paper overall.
The paper provides both theoretical arguments and empirical evidence that was deemed sufficient by all reviewers.